# CLIMATEAR: MULTI-SCALE AUTOREGRESSIVE GENERATIVE MODELING FOR SEASONAL-TO-INTERANNUAL CLIMATE FORECASTING

## ABSTRACT

Accurate Seasonal-to-interannual climate forecasting provides critical support for decision-making in agriculture, energy, and disaster preparedness. Current deterministic models often fail to capture climate uncertainty, while existing generative approaches oversimplify the system by neglecting key spatiotemporal dependencies and cross-scale interactions. To address these limitations, we introduce **ClimateAR**, an AutoRegressive generative model for probabilistic Climate forecasting. The framework incorporates two novel components: (1) an aligned tokenizer that bridges and aligns heterogeneous simulation and real-world data to improve transferability across domains, and (2) a mixed-scale conditioning mechanism that captures multi-scale climate interactions for robust probabilistic forecasting. Extensive evaluations on the ERA5 reanalysis dataset show that ClimateAR achieves state-of-the-art performance, improving anomaly correlation skill by 29.27% on average compared to leading baselines. Code is available at
https://anonymous.4open.science/r/ClimateAR-956D.

## 1 INTRODUCTION

Seasonal-to-interannual (S2I) Climate forecasting aims to predict the evolution of the climate system from several months to roughly several years timescales. It provides critical guidance for a wide range of applications, including resource allocation, energy management, and agricultural planning (Meehl et al., 2021). In contrast to weather forecasting, climate prediction focuses on statistical features of anomaly fields over extended periods, which are intrinsically uncertain due to the complex, nonlinear, and chaotic dynamics of the Earth system. As the forecast horizon extends beyond the weather scale (e.g., >1 month), the sources of predictability of the climate system undergo a fundamental transition: short-term, grid-resolved meteorological signals diminish in influence, giving way to slower-evolving modes of internal climate variability (Trenberth et al., 2007). A prime example is the El Niño–Southern Oscillation (ENSO), whose evolution is governed by nonlinear ocean-atmosphere interactions and modulated by stochastic atmospheric transients (Amaya et al., 2025), leading to significant irreducible uncertainty. Thus, probabilistic forecasting offers more realistic descriptions of inherent uncertainty in climate systems and thus improves the predictive skill for the possibilities of occurrence of extreme events (Price et al., 2025).

Over the past decades, Numerical Weather Prediction (NWP) models have significantly advanced climate and weather forecasting by numerically solving the governing equations of atmospheric and oceanic dynamics (Hurrell et al., 2013; Zhang et al., 2019). These physics-based systems offer strong interpretability and long-term stability, rooted in first principles. However, they face inherent trade-offs between computational cost and spatial resolution, which constrain their ability to scale to high-resolution global simulations—particularly for ensemble or real-time applications (Guo et al., 2025). Recently, data-driven approaches based on deep learning have emerged as a promising alternative (Hwang et al., 2019; Lam et al., 2023; Nguyen et al., 2023; Bi et al., 2023; Liu et al., 2025). By learning complex spatiotemporal patterns directly from reanalysis datasets, they achieve greater computational efficiency and good accuracy in short- to medium-range forecasting. Nevertheless, most existing deep learning methods operate at the pixel (or grid) level and produce deterministic outputs, thereby failing to capture the probabilistic nature of climate variability and underrepresenting uncertainty in long-range predictions.

To address this gap, probabilistic methods based on generative adversarial networks (GANs) (Ravuri et al., 2021; Zhang et al., 2023) and diffusion models (Price et al., 2025; Brenowitz et al., 2025) have been introduced. These methods explicitly represent uncertainty by modeling the joint distribution of climate variables, enabling ensemble predictions. Yet, they frequently treat the climate system as an oversimplified monolithic stochastic process, potentially overlooking essential multi-scale spatiotemporal correlations. For example, they may fail to capture the energy cascade from large-scale climate modes (e.g., ENSO) to regional weather anomalies, a key mechanism governing seasonal predictability (Amaya et al., 2025).

In this paper, we explore the visual autoregressive (AR) model (Tian et al., 2024; Xiong et al., 2024; Li et al., 2024; Han et al., 2025) for the probabilistic modeling of S2I climate forecasting. By representing images as sequences of discrete tokens across multiple scales, AR models capture rich semantic and spatial dependencies while avoiding oversimplified stochastic assumptions. Crucially, the multiscale inductive bias is particularly useful to model climate systems, where phenomena such as teleconnections (i.e., large-scale and statistically significant relationships between climate anomalies in geographically distant regions) and scale interactions are fundamental. In addition, the token classification-based generative process is naturally aligned with climate forecasting, where accuracy and physical consistency are prioritized over diversity.

Despite its theoretical promise, adapting the visual AR model to climate forecasting faces two major challenges: (1) **Highly heterogeneous meteorological data.** Due to data limitations, a common training paradigm in climate research (Nguyen et al., 2023) involves pre-training on simulated data (e.g., CMIP6 (Eyring et al., 2016a)) and fine-tuning on reanalysis data (e.g., ERA5 (Hersbach et al., 2023)). However, the distribution discrepancy between the simulated and reanalysis data hinders effective model transfer. Standard visual tokenizers in AR models treat inputs as homogeneous channels (Esser et al., 2021), lacking mechanisms to align semantic representations across domains. (2) **Complex conditional modeling.** Climate forecasts depend on accurately conditioning the generative process on the prior climatic state—a high-dimensional, multi-variable field with extremely high information density. Effectively integrating such complex conditions is challenging, particularly when capturing cross-scale interactions that govern large-scale predictability in climate dynamics, such as the teleconnections. In contrast, typical existing generative conditioning mechanisms, designed for low-dimensional inputs like text prompts, lack the capacity to handle teleconnection and fail to effectively inject complex climatic constraints (Pang et al., 2024).

To overcome these challenges, we propose **ClimateAR**, the first AutoRegressive model designed for probabilistic Climate forecasting. ClimateAR incorporates a novel tokenization strategy and a mixed-scale conditioning mechanism to handle heterogeneous, high-dimensional climate data. The main contributions of ClimateAR are as follows:

- A principled **generative approach** to probabilistic climate forecasting. ClimateAR models the climate system as a multi-scale stochastic process, enabling the generation of ensemble forecasts that capture inherent climate uncertainty and provide a realistic description of climate processes.

- An **aligned tokenizer** that adopts vector quantization with segmented codebooks to represent high-dimensional climate variables efficiently. A shallow-separation and deep-sharing architecture aligns token semantics across simulated and real-world datasets, enhancing transferability.

- A **mixed-scale conditional control** mechanism that combines scale-specific local guidance with a hybrid-scale global prompt, capturing cross-scale interactions (e.g., the influence of large-scale oceanic anomalies on regional temperature) and leveraging high-information-density conditions effectively.

- Extensive experiments on ERA5 reanalysis data demonstrate that ClimateAR outperforms state-of-the-art AI and physical baselines, achieving an average 29.27% improvement in correlation skill. Notably, it exhibits strong performance in forecasting El Niño–Southern Oscillation (ENSO) indices, highlighting its ability to capture essential climate modes.

## 2 RELATE WORK

Deep learning methods have demonstrated significant advantages over traditional numerical approaches for weather and climate forecasting (Ren et al., 2021; Chen et al., 2023b; Shi et al., 2025). Recent advances leverage sophisticated models to capture more representative meteorological signals, leading to improved accuracy. For example, Pangu (Bi et al., 2023), ClimaX (Nguyen et al., 2023), and Aurora (Bodnar et al., 2025) employ vision Transformers, while GraphCast (Lam et al., 2023) and Oneforecast (Gao et al., 2025) utilize GNNs. In addition, Fuxi (Chen et al., 2023a) and FuXi-S2S (Chen et al., 2024) leverage FNOs. Beyond purely data-driven models, some deep learning approaches also attempt to incorporate physical knowledge to better capture the underlying dynamics of the climate system. For example, ClimODE (Verma et al., 2024) integrates conservation equations, SFNO (Bonev et al., 2023) adopts the spherical characteristics of Earth, and WeatherGFT (Xu et al., 2024) uses PDE kernels. However, these models typically focus on capturing pixel-level patterns and produce deterministic forecasts, often neglecting the inherent uncertainty of the climate system. To address this gap, several approaches have explored generative models for stochasticity modeling. Notable examples include GANs-based methods (cDCGAN (Sha et al., 2024) and cGAN (Rampal et al., 2024)), and models based on diffusion processes (Graph-EFM (Oskarsson et al., 2024), GenCast (Price et al., 2025), and GenAI (Lopez-Gomez et al., 2025)). However, these methods often treat the climate system as an overly simplified stochastic process, potentially overlooking critical multi-scale spatiotemporal correlations.

## 3 PRELIMINARY

**Problem Definition.** Following standard climate forecasting settings (Meehl et al., 2021; Arias et al., 2021), we adopt monthly-averaged meteorological data as both input and output of the model. The input is an initial climate state $\mathbf{X}_t \in \mathbb{R}^{C \times H \times W}$ at time step $t$, where $C$ represents the number of meteorological variables, $H$ and $W$ represent the height and width of the global latitude-longitude grid, respectively. For the generative model, the forecasting task is to model the probabilistic distribution of all variables at the subsequent time step:

$$p(\mathbf{X}_{t+1} \mid \mathbf{X}_t) = \mathcal{F}(\mathbf{X}_t, \Theta), \tag{1}$$

where $\mathcal{F}$ represents the neural network, $\Theta$ is the learnable parameters of $\mathcal{F}$. Then, long-term forecasts $\mathbf{X}_{t+T}$ can be obtained through a $T$-step iterative forecasting process.

**Simulation to Real-World Transfer.** Existing works have demonstrated that pre-training models with simulated data can significantly enhance performance (Nguyen et al., 2023). Due to the limited availability of long-term observational and reanalysis data, it is challenging to train deep climate models from scratch with sufficient generalization ability. To overcome this data scarcity, we leverage large-scale simulated data from the Coupled Model Intercomparison Project Phase 6 (CMIP6) (Eyring et al., 2016a) during pre-training. This allows the model to learn robust physical patterns and climate dynamics, providing a strong foundational prior. Subsequently, we transfer the model to real-world forecasting by fine-tuning on the ERA5 dataset (Hersbach et al., 2023), augmented with ocean variables from Ocean Reanalysis System 5 (ORAS5) (Zuo et al., 2019) to enhance physical consistency. This simulated-to-real transfer strategy effectively bridges the distribution gap while capitalizing on the complementary strengths of simulated and observed climate data. Thus, our model can integrate diverse variables and possess robust cross-domain alignment capabilities between different data sources.

**Visual Autoregressive Modeling.** Tian et al. (2024) proposed a novelty paradigm named VAR for visual autoregressive modeling. The tokenizer first encodes the input image $\mathbf{I} \in \mathbb{R}^{H \times W \times C}$ into a continuous feature map $\mathbf{f} \in \mathbb{R}^{h \times w \times d}$, and then quantizes it into $K$ multi-scale residual discrete token maps $(\mathbf{r}_1, \mathbf{r}_2, \ldots, \mathbf{r}_K)$ with their corresponding resolution $(h_1, w_1) < (h_2, w_2) < \cdots < (h_K, w_K) = (h, w)$:

$$\mathbf{f} = \varepsilon(\mathbf{I}), \quad \mathbf{res}_k = \mathbf{f} - \sum_{i=1}^{k-1} \mathrm{up}(\mathbf{r}_i, (h, w)), \quad \mathbf{r}_k = \underset{q \in \{1, \ldots, V\}}{\arg\min} \|\mathbf{res}_k - \mathbf{c}_q\|_2, \tag{2}$$

where $\varepsilon$ is an encoder, $\mathbf{res}_k$ is the residual feature map before quantization, $\mathbf{c}_q$ represents the $q$-th vector in the codebook with $V$ capacity, and $\mathrm{up}(\cdot)$ denotes an upsampling operation. Then, the

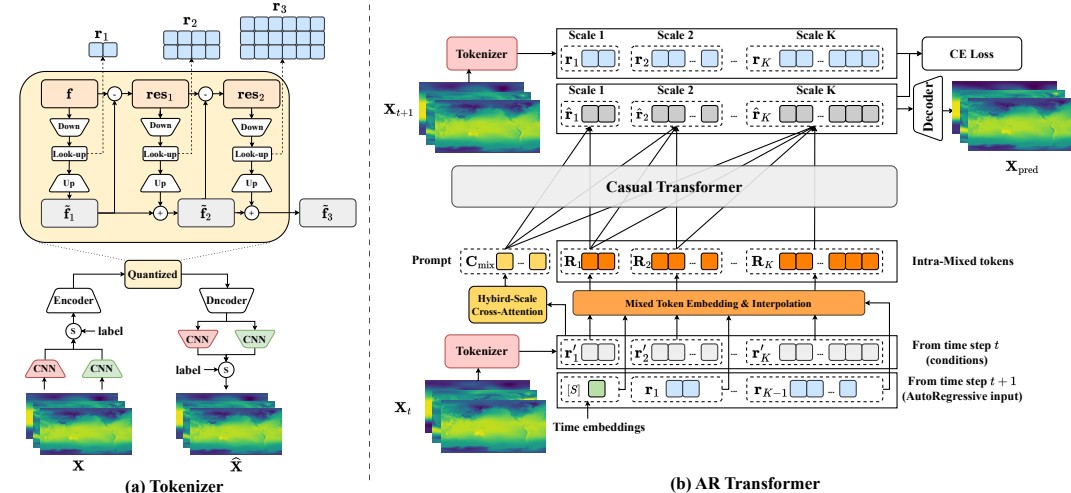

(a) Tokenizer  (b) AR Transformer

Figure 1: The architecture of ClimateAR consists of two key components: (a) a VQ multi-scale tokenizer where **S** replaces a selection operation by label of data source. (b) a decoder-only autoregressive transformer where **CE** represent the cross-entropy loss function.

feature map $\tilde{\mathbf{f}}_k$ at $k$-th scale and the reconstructed image can be expressed as:

$$\tilde{\mathbf{f}}_k = \sum_{i=1}^{k} \text{up}(\mathbf{r}_i, (h, w)), \quad \hat{\mathbf{I}} = \mathcal{D}(\tilde{\mathbf{f}}), \tag{3}$$

where $\tilde{\mathbf{f}} = \tilde{\mathbf{f}}_K$ is the feature map after quantized, $\mathcal{D}$ is a decoder, and $\hat{\mathbf{I}}$ is the reconstructed image. Subsequently, VAR redefines the paradigm of visual autoregressive modeling via the next-scale token map prediction:

$$p(\mathbf{r}_1, \mathbf{r}_2, ..., \mathbf{r}_K) = \prod_{k=1}^{K} p(\mathbf{r}_k \mid \mathbf{r}_{<k}), \tag{4}$$

where $\mathbf{r}_{<k}$ represents token maps prior to scale $k$.

## 4 METHODOLOGY

As illustrated in Fig. 1, ClimateAR consists of a tokenizer and an autoregressive transformer. Given an initial climate state $\mathbf{X}_t$, the tokenizer first encodes it into a sequence of multi-scale token maps $(\mathbf{r}'_1, \mathbf{r}'_2, ..., \mathbf{r}'_K)$, which serve as conditional inputs. The autoregressive transformer then generates the forecast iteratively: starting from a learnable start token [**S**], it produces subsequent token maps scale by scale, with each step conditioned on previously generated tokens and the corresponding scale-specific features. To further guide the generation process globally, the conditional token maps are compressed into a hybrid-scale prompt that acts as a prefix across all scales. Finally, the decoder reconstructs the future climate state $\mathbf{X}_{t+1}$ from the full set of predicted token maps.

### 4.1 TOKENIZE HETEROGENEOUS METEOROLOGICAL DATA

**High-Dimensional Feature Extraction.** Following vector-quantized (VQ) methods, we encode multivariate climate data $\mathbf{X}_t \in \mathbb{R}^{C \times H \times W}$ into a sequence of multi-scale discrete token maps $(\mathbf{r}_1, \mathbf{r}_2, ..., \mathbf{r}_K)$. Given the high dimensionality and intense information of typical meteorological variables, we adopt a partitioned quantization strategy to enhance codebook capacity and reduce latent feature dimension. Specifically, the residual feature $\mathbf{res}_k \in \mathbb{R}^{h \times w \times d}$ at scale $k$ is split along the channel dimension into $N$ disjoint segments, and each of them is quantized independently using a separate codebook of size $V$:

$$\mathbf{res}_k = [\mathbf{res}_k^{(1)}, \mathbf{res}_k^{(2)}, \ldots, \mathbf{res}_k^{(N)}], \quad \mathbf{res}_k^{(n)} \in \mathbb{R}^{h \times w \times d_n} \tag{5}$$

$$\mathbf{r}_k^{(n)} = \underset{q \in \{1, ..., V\}}{\arg\min} \left\| \mathbf{res}_k^{(n)} - \mathbf{c}_q^{(n)} \right\|_2, \tag{6}$$

where $\sum_{n=1}^{N} d_n = d$. This partition increases the effective codebook capacity from $V$ to $V^N$ while reducing the per-segment feature dimension to $d/N$, making the model both expressive and computationally tractable. Each token $\mathbf{r}_k^{(i,j)}$ is thus represented by $N$ disjoint labels during autoregressive training.

**Cross-Domain Alignment.** To enable knowledge transfer from simulation-based pre-training to real-world forecasting, we design an **aligned VQ tokenizer** that projects heterogeneous climatic data—from simulated and real-world sources—into a shared semantic space. As illustrated in Fig. 1, we employ domain-specific convolutional layers in the initial stages of the encoder and the final stages of the decoder to handle low-level distribution discrepancy, while sharing deep network layers and codebooks across domains to ensure high-level climate patterns are encoded consistently. This "shallow-separation, deep-alignment" design promotes learning transferable and domain-invariant representations, improving generalization when fine-tuning on real observations.

## 4.2 CONDITIONAL CONTROL

To effectively incorporate multi-scale climatic dependencies into the generative process, we condition the autoregressive generation with multi-scale information from the previous state. Specifically, the tokenizer encodes both the initial state $\mathbf{X}_t$ and the target state $\mathbf{X}_{t+1}$ into multi-scale token sequences $(\mathbf{r}_1', \mathbf{r}_2', ..., \mathbf{r}_K')$ and $(\mathbf{r}_1, \mathbf{r}_2, ..., \mathbf{r}_K)$, respectively. The conditional distribution $p(\mathbf{X}_{t+1} \mid \mathbf{X}_t)$ can be expressed autoregressively over scales:

$$p(\mathbf{X}_{t+1} \mid \mathbf{X}_t) = p(\mathbf{r}_1, \mathbf{r}_2, ..., \mathbf{r}_K \mid \mathbf{r}_1', \mathbf{r}_2', ..., \mathbf{r}_K') = \prod_{k=1}^{K} p(\mathbf{r}_k \mid \mathbf{r}_{<k}, \mathbf{r}_{\leq K}'). \tag{7}$$

**Intra-scale Mixed Token.** Directly conditioning on all scales poses convergence challenges due to the high information density of climate states. We therefore introduce a *intra-scale mixed-token* mechanism that fuses the autoregressive feature $\tilde{\mathbf{f}}_{k-1}$ (reconstructed from tokens $\mathbf{r}_{\leq k-1}$ through Eq. 3) and with the conditional feature $\tilde{\mathbf{f}}_k'$ from $\mathbf{r}_{\leq k}$ at each scale. This allows the model to approximate the conditional distribution as:

$$\mathbf{R}_k = \text{Concat}\Big(\text{down}(\tilde{\mathbf{f}}_{k-1}, (w_k, h_k)), \text{down}(\tilde{\mathbf{f}}_k', (w_k, h_k))\Big), \tag{8}$$

$$p(\mathbf{r}_k \mid \mathbf{r}_{<k}, \mathbf{r}_{\leq K}') \approx p(\mathbf{r}_k \mid \mathbf{R}_{\leq k}), \tag{9}$$

where $\text{down}(\cdot)$ performs spatial interpolation to match the target token resolution and $\mathbf{R}_k$ represents the mixed token at scale $k$. This approximation effectively maintains scale-wise physical consistency while leveraging the efficiency of a standard autoregressive framework.

**Hybrid-Scale Prompt.** While the intra-scale mixed token aligns conditional information at corresponding scales, the autoregressive process inherently lacks access to subsequent finer-scales $(\mathbf{r}_{>k}')$ conditional information. This presents a significant limitation for climate forecasting, as the evolution of large-scale climate phenomena (e.g., ENSO) is often modulated by finer-scale processes (e.g., regional convective activity or oceanic eddies). To capture these interactions, we propose a hybrid-scale prompt $\mathbf{C}_{\text{mix}}$ that compresses global, multi-scale information from the entire conditional token sequence into a global context using cross-attention:

$$\mathbf{C}_{\text{mix}} = \text{Attention}(q = \mathbf{q}, kv = (\mathbf{r}_1', \mathbf{r}_2', ..., \mathbf{r}_K')), \tag{10}$$

where $\mathbf{q}$ is a learnable hybrid-scale query sequence. The resulting representation serves as a prefix to the autoregressive sequence, enabling full cross-scale awareness. It is important to note that the compressed $\mathbf{C}_{mix}$ resides in a continuous space, whereas $(\mathbf{r}_1', \mathbf{r}_2', ..., \mathbf{r}_K')$ are discrete tokens. Therefore, even though $\mathbf{C}_{mix}$ is much shorter, we can still achieve nearly lossless compression of hybrid-scale information. The exact conditional generation thus becomes:

$$p(\mathbf{r}_k \mid \mathbf{r}_{<k}, \mathbf{r}_{\leq K}') = p(\mathbf{r}_k \mid \mathbf{C}_{\text{mix}}, \mathbf{R}_{\leq k}). \tag{11}$$

Finally, we apply a decoder-only transformer to the entire token maps $[\mathbf{C}_{mix}, \mathbf{R}_1, \mathbf{R}_2, ..., \mathbf{R}_k]$ to obtain next-scale token prediction $\mathbf{r}_k$.

**Noise-Augmented Teacher-Forcing.** Autoregressive models trained with teacher forcing are prone to *exposure bias*: during training, the model uses ground-truth tokens as input, while during inference it must use its own predictions, leading to error accumulation. To mitigate this issue, during

training, we stochastically replace ground-truth tokens with random tokens during training while preserving the residual structure across scales:

$$\mathbf{res}_k = \mathbf{f} - \sum_{i=1}^{k-1} \mathrm{up}(\mathbf{r}_i^{\mathrm{noisy}}, (h, w)), \ \mathbf{r}_k = \underset{q \in \{1,\dots,V\}}{\arg\min} \|\mathbf{res}_k - \mathbf{c}_q\|_2, \tag{12}$$

$$\mathbf{r}_k^{\mathrm{noisy}} = \mathbf{r}_k \odot \mathbf{M} + \mathbf{r}_q \odot (1 - \mathbf{M}), \tag{13}$$

where $\mathbf{M}_{i,j} \sim B(1, p)$ is a mask for whether the element of $\mathbf{r}_k^{i,j}$ should be replaced by a random token $\mathbf{r}_q^{i,j}$, $\odot$ represents Hadamard product, and more details are in Appendix A.1. Introducing such noise in autoregressive conditions narrows the gap between training and inference, and forces the model to rely more heavily on the conditional state $\mathbf{X}_t$ to reconstruct the target state, thus enhancing its ability to correct the accumulated errors.

### 4.3 TRAINING PROCEDURE

**VQ-VAE Training:** We first train the VQ-VAE on a mixture of real-world (ERA5, ORAS5) and simulated (CMIP6) climate data. Domain indicators are included to facilitate the cross-domain alignment described in Section 4.1. Once trained, the VQ-VAE weights are frozen, and the model serves as a fixed tokenizer for encoding input states and decoding predicted tokens. The training objective combines VQ loss with perceptual terms:

$$\mathcal{L} = \left\|\mathbf{X} - \hat{\mathbf{X}}\right\|_2^2 + \sum_{k=1}^{K} \|\mathbf{res}_k - \mathbf{r}_k\|_2^2 + \lambda_s \mathcal{L}_s(\mathbf{X}, \hat{\mathbf{X}}), \tag{14}$$

where $\hat{\mathbf{X}}$ are the reconstructed data, $\mathcal{L}_s$ is defined as $\mathcal{L}_s(\mathbf{X}, \hat{\mathbf{X}}) = 1 - \mathrm{SSIM}(\mathbf{X}, \hat{\mathbf{X}})$, denoting the structural similarity index measure to preserve large-scale patterns, and $\lambda_s$ controls its weight.

**AR Pretraining:** We pretrain the autoregressive transformer on CMIP6 simulations to learn fundamental climate evolution patterns. At this stage, the model learns to predict the next-scale token maps conditioned on previous tokens and the hybrid-scale prompt. Since the tokenizer discretizes the climate state, we formulate the task as a classification problem, optimizing the cross-entropy loss $\mathcal{L}_{\mathrm{CE}}$ over the codebook indices instead of using pixel-level regression.

**AR Fine-tuning:** Finally, we fine-tune the pre-trained autoregressive model on the ERA5 and ORAS5 reanalysis datasets to adapt it to real-world forecasting, utilizing the same cross-entropy objective. After fine-tuning, the model performs iterative forecasting by using each single-step forecast as the input for the subsequent time step, enabling seamless extension to long-term forecasts.

## 5 EXPERIMENTS

### 5.1 EXPERIMENTAL SETUP

**Datasets.** We pretrain ClimateAR using historical simulations from the Community Earth System Model version 2 (CESM2) (Danabasoglu et al., 2012), which contributed to the Coupled Model Intercomparison Project Phase 6 (CMIP6) (Eyring et al., 2016b). The training data consist of nine ensemble members spanning 1850–2014, interpolated to a $1° \times 1°$ latitude–longitude grid. Model fine-tuning and evaluation are performed on the ERA5 atmospheric reanalysis and ORAS5 ocean reanalysis datasets (Zuo et al., 2019). The input variables include 4 atmospheric fields at 4 vertical levels, 6 surface variables, and 2 oceanic variables at 6 vertical levels, totaling 34 channels. (See Appendix A.2 for the full variable list.) Missing oceanic values over land are filled with the global mean and masked during training. We use CMIP6 data (1850–2014) for pre-training, ERA5 and ORAS5 (1958–1999) for fine-tuning, 2000–2004 for validation, and 2005–2014 for testing.

**Baselines.** We evaluate ClimateAR against four leading data-driven methods: Pangu (Bi et al., 2023), GraphCast (Lam et al., 2023), Oneforecast (Gao et al., 2025), and ClimaX (Nguyen et al., 2023). Pangu and ClimaX are based on Vision Transformers (ViT) (Dosovitskiy et al., 2021), while Oneforecast and GraphCast employ graph neural networks. For comparison with conventional physics-based approaches, we include seasonal predictions from the German Meteorological

Service (DWD) (Paxian et al., 2023). The DWD system uses a fully coupled climate model, initialized by nudging atmospheric and oceanic components toward reanalysis states. It produces monthly forecasts with lead times up to six months, evaluated here over the hindcast period 1993–2016. Further baseline details are provided in Appendix A.2.2.

**Metrics.** To evaluate the forecasting skill on S2I timescales, we use the latitude-weighted Anomalous Correlation Coefficient (ACC) and Root Mean Square Error (RMSE). RMSE measures the absolute deviation in predicted values, providing insight into the overall grid-wise precision. Crucially, the ACC is employed as the **principal metric** for assessing the forecasting skill. It quantifies the spatial and temporal agreement between predicted and observed anomalies—deviations from the long-term climatological mean, thus directly measuring the skill in forecasting meaningful climate variability (e.g., the development of ENSO or blocking events). By focusing on the pattern similarity of anomalies, the ACC is more robust to systematic biases in the mean state than RMSE, as minimizing RMSE tends to make the prediction close to the mean but ignore the anomalies, which is important in climate forecasting.

$$\text{ACC}(\nu) = \sum_{h,w} \frac{\alpha(h)}{HW} \frac{\sum_t \hat{\mathbf{A}}_{\nu,h,w,t} \mathbf{A}_{\nu,h,w,t}}{\sqrt{\sum_t \hat{\mathbf{A}}_{\nu,h,w,t}^2 \sum_t \mathbf{A}_{\nu,h,w,t}^2}}, \tag{15}$$

$$\text{RMSE}(\nu) = \sum_{h,w} \frac{\alpha(h)}{HW} \sqrt{\frac{1}{T} \sum_t (\hat{\mathbf{X}}_{\nu,h,w,t} - \mathbf{X}_{\nu,h,w,t})^2}, \tag{16}$$

where $\alpha(h) = H\cos(\lambda_h)/\sum_i \cos(\lambda_i)$ is the latitude weight factor, $\mathbf{X}_{\nu,h,w,t}$ is the ground truth for variable $\nu$ at time step $t$ in pixel $(h, w)$ and $\hat{\mathbf{X}}_{\nu,h,w,t}$ is its prediction. The anomaly is $\mathbf{A}_{\nu,h,w,t} = (\mathbf{X}_{\nu,h,w,t} - \mathbf{C}_{\nu,t})$, where $\mathbf{C}_{\nu,t}$ is the climatology (i.e., the mean of variable $\nu$ at the time with $t$ in long-term historical periods).

**Implementation Details.** We configure ClimateAR with a batch size of 64, 8 codebooks with a size of 4096, a hidden dimension of 1024 for both the VQ-VAE and Transformer, 16 heads and 16 layers in the Transformer architecture, and a learning rate of 0.0005 scheduled via linear warmup and cosine annealing. For the baselines, we retrained the model on monthly data with default hyperparameters to adapt to the monthly-scale data prediction task. All these models are trained on 8 NVIDIA A800 80G GPUs. For a fair comparison, all models are pre-trained for a maximum of 50 epochs and fine-tuned for a maximum of 20 epochs with an early stop strategy. More implementation details are provided in Appendix A.2.3.

## 5.2 OVERALL PERFORMANCE

**Baseline Comparison.** We evaluate ClimateAR against baseline models on 12 key variables spanning multiple pressure levels and surface fields, including geopotential height (z500, z850, z1000), temperature (t2m, t500, t850, t1000), sea surface temperature (sst), precipitation rate (pr), sea level pressure (psl), and surface wind components (u1000, v1000). As summarized in Fig. 2 and detailed in Appendix A.4.1, ClimateAR achieves superior performance across most variables and lead times, with an average ACC improvement of 29.27% over the strongest baseline. More comparison (RMSE comparison in Fig. 8 and low latitude regional forecasting from 1 to 14-month in Fig. 9) can be found in Appendix. A.4.1. This demonstrates the advantage of leveraging high-level, tokenized representations learned from heterogeneous data in guiding climate forecasts.

Notably, ClimateAR exhibits the most significant gains in predicting near-surface temperatures (t2m and t1000)—variables strongly influenced by large-scale climate modes such as ENSO. These results suggest that the model's discrete token-based formulation effectively captures climate-mode dependencies, moving beyond pixel-level regression. While Pangu and ClimaX deliver competitive accuracy through ViT-based global context modeling, their deterministic nature limits their ability to represent systemic climate uncertainty. ClimateAR addresses this by explicitly modeling stochasticity through its autoregressive generative framework.

**Visualization.** Spatial distributions of ACC for sea surface temperature (Fig. 3) reveal that all models exhibit higher skill in tropical oceans, where climate variability is more predictable. ClimateAR consistently outperforms baselines in these regions, particularly over the tropical Pacific, indicating its enhanced capacity to capture dominant modes of interannual variability. This advantage stems

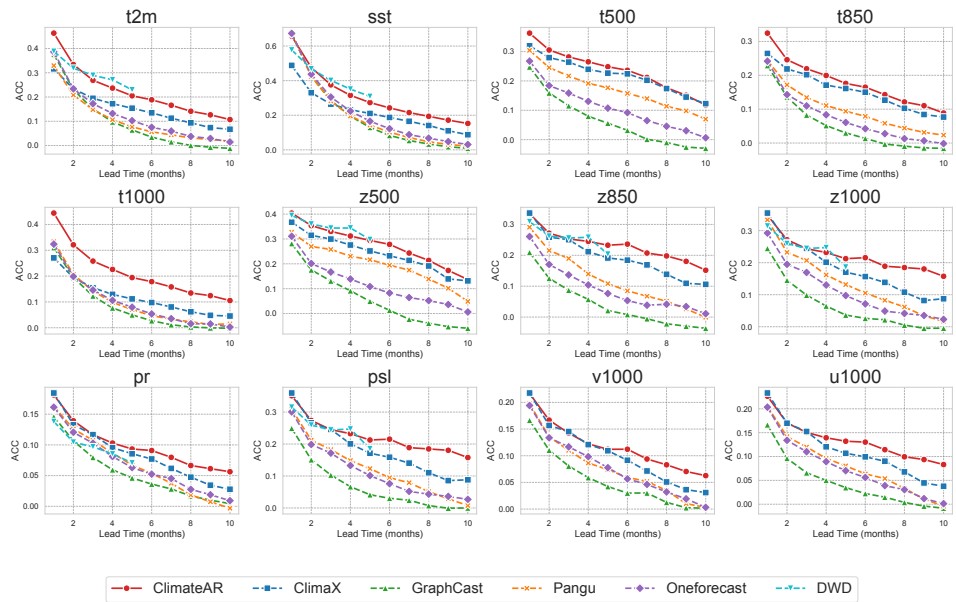

Figure 2: Comparison of ACC for ClimateAR and baselines in the global forecast from 1 to 10-month lead times. More details can be found in Appendix A.4.1.

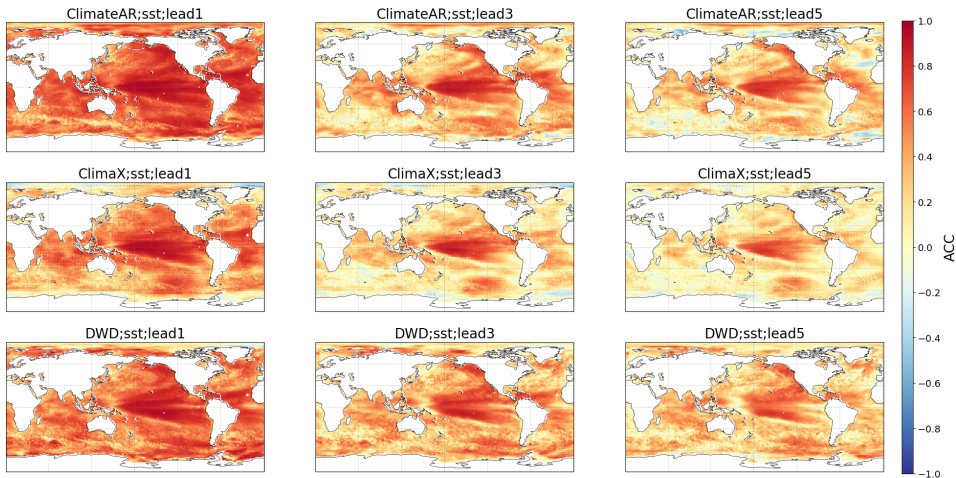

Figure 3: The global ACC distribution of sea surface temperature forecasting with 1-, 3-, and 5-month lead times in the testing set.

from the model's ability to represent spatio-temporal dependencies and uncertainty in a unified generative process. Additional visualizations, including RMSE maps, are provided in Appendix A.4.2.

**Zero-shot Forecasting.** To evaluate the transferability and generalization capability of ClimateAR, we conduct a zero-shot test on ERA5 data using the pre-trained model trained on the CIMP6 simulated data. From Table 1, we can observe that ClimateAR outperforms all the data-driven models. These results demonstrate its robust transferability and strong generalization performance.

## 5.3 ENSO FORECASTING

For S2I climate forecasting tasks, the El Niño–Southern Oscillation (ENSO) phenomenon is a widely monitored indicator. The performance for ENSO forecasting can be evaluated by the ENSO index calculated by the mean results of t2m in the Niño 3.4 region ($170°$W to $190°$E and $5°$S to $5°$N). In Fig. 4, we compare the prediction with the observations, which demonstrates that ClimateAR can predict the ENSO index with high accuracy up to 10 months in advance. Specifically, for example, in the test set, a pronounced El Niño event occurred in 2009, and ClimateAR is able to effectively

Table 1: Zero-shot forecasting mean results with lead times of months 1-6 of ClimateAR and data-driven baselines. All the models are only pre-trained on simulated datasets. The best results are bolded, and the second-best results are underlined.

| | RMSE (↓) | | | | | ACC (↑) | | | | |
|---|---|---|---|---|---|---|---|---|---|---|
| | ClimateAR | GraphCast | Pangu | Oneforecast | ClimaX | ClimateAR | GraphCast | Pangu | Oneforecast | ClimaX |
| z500 | **35.333** | 42.493 | 41.263 | 40.663 | 38.860 | **0.323** | 0.129 | 0.240 | 0.181 | 0.295 |
| t2m | **2.011** | 2.032 | 2.269 | 2.032 | 2.039 | **0.276** | 0.144 | 0.134 | 0.167 | 0.168 |
| pr($\times 10^{-5}$) | 2.037 | 2.041 | **1.986** | 2.053 | 2.003 | **0.112** | 0.068 | 0.091 | 0.088 | 0.094 |
| psl | **285.299** | 306.048 | 292.689 | 303.579 | 301.670 | **0.238** | 0.094 | 0.155 | 0.158 | 0.205 |

forecast its onset and decay. In addition, we also compare the ACC between ClimateAR and the baselines for the ENSO index prediction task in Fig. 4. We can observe that ClimateAR outperforms all the baselines, which demonstrates its ability to learn potential climate mode patterns.

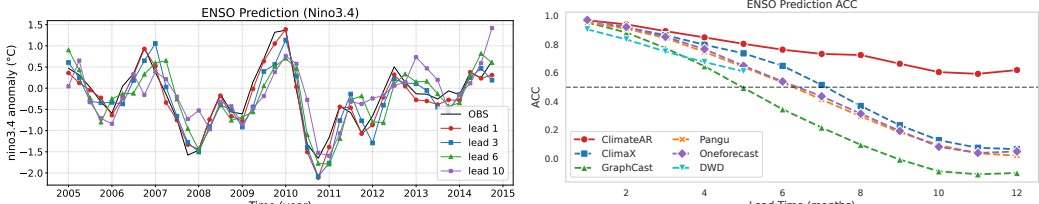

Figure 4: The ENSO index forecast results. The left figure is a comparison of ClimateAR's three-month average ENSO index forecast results and observations at different lead times. The right figure shows the ACC against the forecasting lead time of the ENSO index for ClimateAR and baselines.

## 5.4 ENSEMBLE FORECASTING

**Baseline Comparison on Ensemble Forecasting.** In Fig. 5, we present Continuous Ranked Probability Score (CRPS) metrics (definition in Appendix. A.3) of 4 key variables for global forecasting tasks. For the deterministic baseline, we obtained different ensemble forecast members through slight perturbations in the initial field. ClimateAR outperforms all baselines with an average CRPS reduction of 23.33%, demonstrating its advantage in probabilistic modeling over the deterministic model.

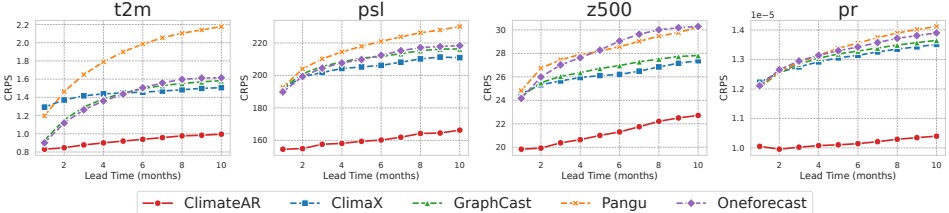

Figure 5: Comparison of CRPS for ClimateAR and baselines in the global forecast from 1 to 10-month lead times. More details can be found in Appendix A.4.1.

**Probabilistic Forecasting on ENSO.** In Fig. 6, we present the performance of the Probabilistic forecasting on the ENSO index, and we find that: (1) Twice the standard deviation of the ensemble forecast can effectively capture the true observations, proving the effectiveness of climateAR in modeling uncertainties of the climate system. (2) Starting from different time steps, within 12 months, most of the ensemble members of ClimateAR are able to effectively predict the development and dissipation of the ENSO phenomenon, demonstrating its effective capture of climate patterns. In addition, we conduct a power spectral density analysis to show the predicted periodicity characteristics of ENSO in Appendix. A.4.3.

## 5.5 DECADAL CLIMATE FORECASTING

To explore the long-term climate simulation potential of ClimateAR, We apply a simple rolling fine-tune procedure (details in Appendix A.5) to mitigates long-term climatology drift and then conduct a decadal forecasting experiment. We initialize the model with the January 1958 climate state

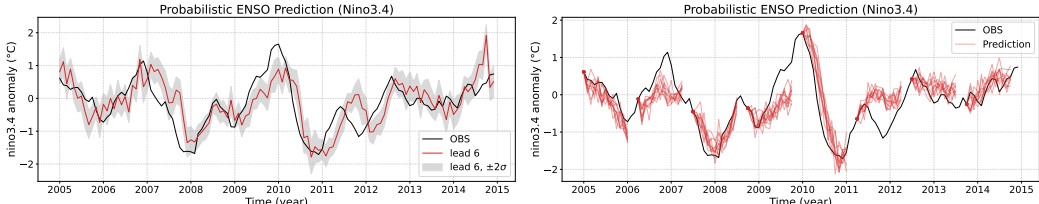

Figure 6: Ensemble forecast results of ENSO index. The left figure compares monthly average forecast results of ClimateAR and observations at 6-month lead time, where the shaded area represents the range indicating twice the ensemble standard deviation. The right figure shows the prediction over 12 months initialized at different start times, including various ensemble forecast members.

and iteratively generate monthly predictions through December 2020, using SST as the only external forcing input. We evaluate the resulting multi-decadal time series by examining the annual and monthly global-mean t2m over 1958–2020 in Fig. 7. ClimateAR produces a stable and physically consistent evolution of global-mean temperature that closely follows the observed long-term trend. This shows that ClimateAR is capable of maintaining coherent large-scale climate statistics over extended periods, highlighting its effectiveness for decadal climate forecasting.

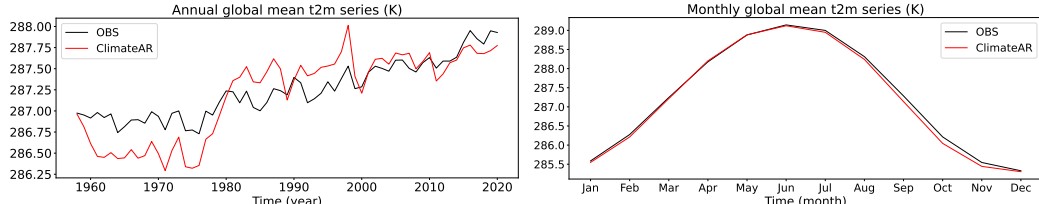

Figure 7: The Global annual-mean and monthly-mean t2m time series of ClimateAR and obversation over 1958–2020.

## 5.6 ABLATION STUDY

**Tokenizer.** We design two variants: (1) Removing the cross-domain alignment module (-w/o align) and using a shared network for different data sources. (2) Removing multiple codebooks and not partitioning the feature vector (-w/o part), using a single codebook for vector quantization.

**Conditional Control.** We design three variants: (1) Removing the hybrid-scale prompt (-w/o hybrid) of token maps. (2) Removing the intra-scale mix tokens (-w/o intra), only using the predicted token as the input to the autoregressive function and using the conditional information as a prefix of token maps. (3) Removing the noise of teach-forcing input (-w/o noise) and using correct tokens as both target and input.

From Table 2 we can find that the lack of any key design will degrade the performance of ClimateAR, which verifies the effectiveness of our proposed methods.

Table 2: Ablation result for the 6 months averaged ACC of ClimateAR.

| | ACC (↑) | | | | | |
|---|---|---|---|---|---|---|
| | | tokenizer | | condition | | |
| | ClimateAR | w/o align | w/o part | w/o prompt | w/o intra | w/o noise |
| z500 | **0.331** | 0.300 | 0.282 | 0.304 | 0.152 | 0.296 |
| t2m | **0.285** | 0.280 | 0.267 | 0.268 | 0.091 | 0.257 |
| pr | **0.121** | 0.116 | 0.120 | 0.107 | 0.021 | 0.097 |
| psl | **0.258** | 0.236 | 0.241 | 0.218 | 0.061 | 0.213 |

## 6 CONCLUSIONS

In this work, we introduce ClimateAR, the first autoregressive probabilistic model for S2I climate forecasting. We introduce novel tokenization and control techniques to extract and utilize high-level features of highly heterogeneous meteorological data for climate forecast generation. Extensive experimental results demonstrate that ClimateAR outperforms all competitive baselines.

## 7 ETHICS STATEMENT

As our work only focuses on the weather and climate forecasting problem, there is no potential ethical risk.

## 8 REPRODUCIBILITY STATEMENT

In the main text, we have formally defined the model architecture with equations. All the implementation details, including dataset descriptions, metrics, and experiment configurations, are provided in the manuscript and the code (available online).

## 9 DECLARATION OF LLM USAGE

The author of this paper only used LLM as a grammar checker and simple text polishing tool. LLM was not used in any of the ideas or technical implementations.

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

## A APPENDIX

### A.1 ADDITIONAL METHOD DETAILS

**Tokenizer alignment and partition.** For our multi-scale VQ tokenizer implementation, we incorporate two key technical components in the tokenizer pre-training procedure: partition with multi-codebook matching and labeled cross-domain data alignment. The detailed implementation algorithm is presented in Alg.1.

---

**Algorithm 1** Aligned tokenizer and partition.

---

**Input:** Input raw data $\mathbf{X}$, label of data source $l$, scale $K$, partitions $N$
    **Selection by label:** $\mathbf{X}_l = \text{CNN}_l(\mathbf{X})$
    $\mathbf{f} = \varepsilon(\mathbf{X}_l)$
    **Initialize:** $\hat{\mathbf{f}} = 0$
    **for** $k = 1$ to $K$ **do**
        $\mathbf{res}_k = (\text{down}(\mathbf{f}, (h_w, h_k)))$
        $[\mathbf{res}_k^{(1)}, \mathbf{res}_k^{(2)}, \ldots, \mathbf{res}_k^{(N)}] = \mathbf{res}_k$
        **for** $i = 1$ to $N$ **do**
            $\mathbf{r}_k^{(i)} = \text{look\_up}(\mathbf{res}_k^{(i)})$
        **end for**
        $\mathbf{r}_k = [\mathbf{r}_k^{(1)}, \mathbf{r}_k^{(2)}, \ldots, \mathbf{r}_k^{(N)}]$
        $\mathbf{f} = \mathbf{f} - \text{up}(\mathbf{r}_k, (h, w))$
        $\hat{\mathbf{f}} = \hat{\mathbf{f}} + \text{up}(\mathbf{r}_k, (h, w))$
        $\hat{\mathbf{f}}_k = \hat{\mathbf{f}}$
    **end for**
    $\hat{\mathbf{X}}_l = \mathcal{D}(\hat{\mathbf{f}})$
    **Selection by label:** $\hat{\mathbf{X}} = \text{CNN}_l(\hat{\mathbf{X}}_l)$
    **return** $\hat{\mathbf{X}}$

---

**Noise Injection.** Furthermore, we introduce noise during the training phase of the autoregressive (AR) model to alleviate exposure bias inherent in autoregressive architectures. The whole process is illustrated in Alg.2.

---

**Algorithm 2** Tokenize data with noise.

---

**Input:** Input feature map $\mathbf{f}$, scale $K$, noise rate $p$
    **Initialize:** target token maps $\mathbf{r}_{tgt} = []$, teach-forcing input token_maps $\mathbf{r}_{in} = []$.
    **for** $k = 1$ to $K$ **do**
        $\mathbf{r}_k = \text{Quantize}(\mathbf{f})$
        $\mathbf{r}_{tgt} = \mathbf{r}_{tgt} \cup \{\mathbf{r}_k\}$
        $\mathbf{r}_k^{\text{noisy}} = \text{Random\_Replace}(\mathbf{r}_k, p)$
        $\mathbf{r}_{tgt} = \mathbf{r}_{tgt} \cup \{\mathbf{r}_k^{\text{noisy}}\}$
        $\mathbf{f} = \mathbf{f} - \text{up}(\mathbf{r}_k, (h, w))$
    **end for**
    **return** $\mathbf{r}_{tgt}, \mathbf{r}_{in}$

---

## A.2 ADDITIONAL EXPERIMENTAL DETAILS

### A.2.1 DATASETS

**CMIP6 datasets.** We use historical simulations from the Community Earth System Model version 2 (CESM2), developed by the National Center for Atmospheric Research (NCAR) (Danabasoglu et al., 2012), which participated in the Coupled Model Intercomparison Project Phase 6 (CMIP6) (Eyring et al., 2016). The CESM2 historical simulations comprise 9 ensemble realizations with different initial conditions, covering 1850–2014, which form 9 simulated datasets used for our model pre-training. All fields were interpolated onto a $1° \times 1°$ grid using bilinear interpolation.

**ERA5 datasets.** The ERA5 dataset is the fifth-generation reanalysis product from the European Centre for Medium-Range Weather Forecasts (ECMWF). ERA5 provides hourly estimates of atmospheric, land, and oceanic climate variables. The data cover the Earth on a 30-kilometer grid and resolve the atmosphere using 137 vertical levels from the surface to an altitude of 80 kilometers. We used the period from 1958 to 2014 for model training. All fields were interpolated onto a $1° \times 1°$ grid using bilinear interpolation.

**ORAS5 datasets.** The ORAS5 dataset is the reanalysis product of the ECMWF OCEAN5 system, which is a new global eddy-permitting ocean-sea ice ensemble reanalysis system. ORAS5 provides ocean state estimates from 1979 to the present and extends historical records back to 1958, assimilating temperature and salinity in-situ observations as well as sea ice concentration data. We used the period from 1958 to 2014 for model training. All fields were interpolated onto a $1° \times 1°$ grid using bilinear interpolation.

**Variables.** Table 3 presents all the variables we use in experiments, including 4 atmospheric variables at 4 vertical levels, 6 single-level variables, and 2 oceanic variables at 6 vertical levels, totaling 34 variables. For sea surface temperature (sst), we use 5-meter sea temperature as an approximation in the corresponding tests.

Table 3: Variables used in experiments.

| Type | Varibale name | Abbreviation | Levels or Depth |
|------|---------------|--------------|-----------------|
| Atmospheric | geopotential | z | 200hPa,500hPa,850hPa,1000hPa |
| Atmospheric | u wind component | u | 200hPa,500hPa,850hPa,1000hPa |
| Atmospheric | v wind component | v | 200hPa,500hPa,850hPa,1000hPa |
| Atmospheric | temperature | t | 200hPa,500hPa,850hPa,1000hPa |
| Ocean | salinity | so | 5m,20m,40m,90m,200m,300m |
| Ocean | sea temperature | temp | 5m,20m,40m,90m,200m,300m |
| Single | 2 metre temperature | t2m | |
| Single | u wind stress component | tauu | |
| Single | v wind stress component | tauv | |
| Single | precipitation rate | pr | |
| Single | sea level pressure | psl | |
| Single | Sea Surface Height | zos | |

### A.2.2 BASELINES

**Pangu.** Pangu is a Vision Transformer-based data-driven model with three-dimensional deep networks equipped with Earth-specific priors. Pangu processes pressure layer variables using 3D patches and single-layer variables using 2D patches, then aggregates the two to perform prediction tasks.

**GraphCast.** GraphCast is a Graph Neural Network-based data-driven model. GraphCast uses multi-mesh method to construct the graph and apply a stage with an encoder, processor, and decoder to learn the complex dynamics of the system.

**Oneforecast.** Oneforecast is a Graph Neural Network-based data-driven model. Oneforecast constructs a multi-scale graph structure and introduces an adaptive messaging mechanism with dynamic gating units for more accurate forecasting.

**ClimaX.** A Vision Transformer-based data-driven foundation model for weather and climate forecasting, which tokenizes variables independently and uses an aggregation module to model the dependency between variables.

**DWD.** We employ seasonal climate forecasts from the German Meteorological Service. The German Meteorological Service forecasting system is based on a fully coupled climate model. Its initialization is achieved by nudging the model's atmospheric and oceanic components toward reanalysis data. The system produces monthly forecasts with a lead time of up to six months, and its hindcast period spans 1993–2016. These datasets are available through the Copernicus Climate Change Service (C3S) hosted by the European Centre for Medium-Range Weather Forecasts (ECMWF).

### A.2.3 ADDITIONAL IMPLEMENTATION DETAILS

**ClimateAR.** In the Table 4, we present the detailed hyperparameter settings of ClimateAR. In addition, our model generates multiple forecast ensemble members through probabilistic sampling over multiple steps, and we ultimately use the average of 200 members as the final forecast result.

Table 4: The detailed hyperparameter settings of ClimateAR

| Hyperparameter | Tokenizer | Hyperparameter | AR Transformer |
|---|---|---|---|
| dropout | 0 | dropout | 0 |
| learning rate ($lr$) | $5e-4$ | pre-train learning rate ($lr_p$) | $5e-4$ |
| batch size ($B$) | 128 | pre-train batch size ($B_p$) | 64 |
| hidden dimension ($d$) | 1024 | hidden dimension ($d$) | 1024 |
| codebooks dimension ($c$) | 128 | noise rate ($p$) | 0.3 |
| number of codebooks ($N$) | 8 | transformer heads ($h$) | 16 |
| size of codebooks ($V$) | 4096 | transformer layers ($l$) | 16 |
| number of scales ($K$) | 10 | number of scales ($K$) | 10 |
| loss function weight ($\lambda_s$) | 1 | fine-tune learning rate ($lr_t$) | $5e-5$ |
| | | fine-tune batch size ($B_t$) | 64 |
| | | prompt length ($L$) | 256 |

**Data-driven baselines.** For Pangu, Oneforecast, and GraphCast, we utilized the source code of One-forecast from `https://github.com/YuanGao-YG/OneForecast`, retraining the model with all default hyperparameters on monthly averaged data and conducting evaluations. For ClimaX, we adopted the publicly available source code from `https://github.com/microsoft/ClimaX`, retraining and evaluating it using the same hyperparameters with $1.40625° \times 1.40625°$ (patch size with $4 \times 4$).

## A.3 CONTINUOUS RANKED PROBABILITY SCORE

The detailed definition of metric CRPS is as follows:

$$\text{CRPS}(F, x_{\text{obs}}) = \int_{-\infty}^{\infty} (F(x) - \text{H}(x - x_{\text{obs}}))^2 \, dx$$

$$H(s) = \begin{cases} 0 & s < 0 \\ 1 & s \geq 0 \end{cases} \tag{17}$$

where $F$ is the cumulative distribution function (CDF) of the prediction $x$, $H$ is the Heaviside function and denotes the CDF of the observation $x_{\text{obs}}$. Then we can compute the global average CRPS using latitude-weighted averaging.

$$\text{CRPS}(\nu) = \sum_{h,w} \frac{\alpha(h)}{THW} \sum_t \text{CRPS}(\mathbf{F}_{\nu,h,w,t}, \mathbf{X}_{\nu,h,w,t}) \tag{18}$$

where $\alpha(h) = H\cos(\lambda_h)/\sum_i \cos(\lambda_i)$ is the latitude weight factor, $\mathbf{X}_{\nu,h,w,t}$ is the ground truth for variable $\nu$ at time step $t$ in pixel $(h, w)$ and $\mathbf{F}_{\nu,h,w,t}$ is the CDF of its prediction.

## A.4 ADDITIONAL RESULTS

### A.4.1 COMPARISON WITH BASELINES

**Comparison on RMSE.** In Fig. 8, we give the comparison of RMSE for ClimateAR and baselines in the global forecast across 1- to 10-month lead times. The full results demonstrate that even though ClimateAR does not rely on pixel-wise MSE loss and only learns token-level classification in a high-level feature space, it still achieves the lowest RMSE prediction error in most cases, proving the effectiveness of modeling systematic uncertainty and spatiotemporal multi-scale dependencies in long-term forecasting tasks.

**Full result.** In Table 5, 6, we present all detailed metrics of the key variables for the global forecasting tasks.

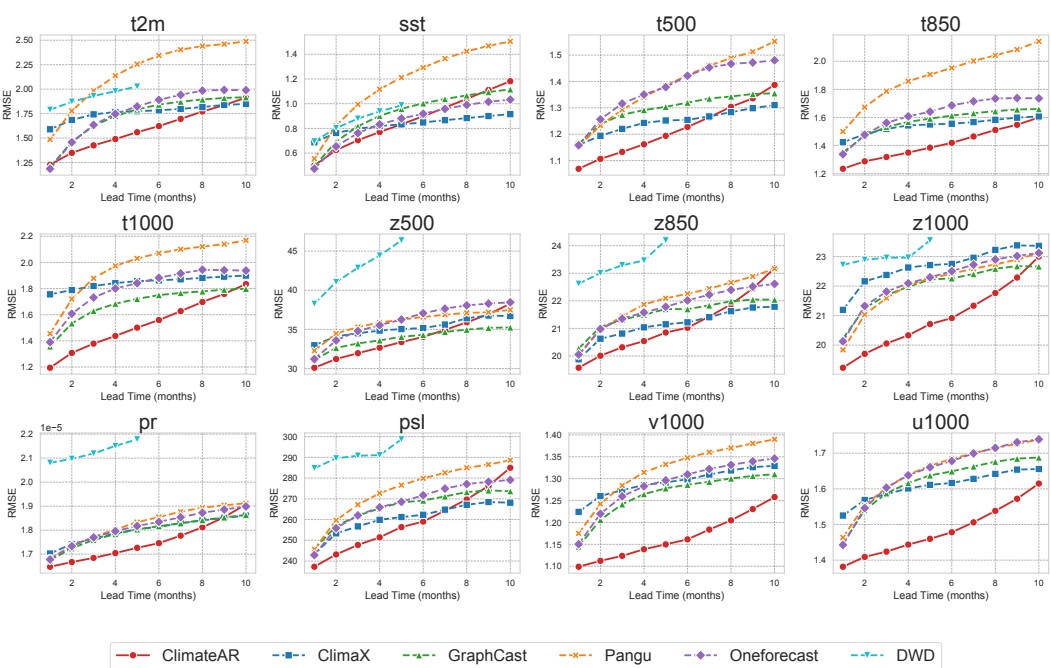

Figure 8: Comparison of RMSE for ClimateAR and baselines in the global forecast across 1- to 10-month lead times

Table 5: Global forecasting CRPS of ClimateAR and baselines. The best results are bolded.

| var | lead | CRPS (↓) ClimateAR | GraphCast | Pangu | Oneforecast | ClimaX | var | lead | CRPS (↓) ClimateAR | GraphCast | Pangu | Oneforecast | ClimaX |
|---|---|---|---|---|---|---|---|---|---|---|---|---|---|
| t2m | 1 | **0.831** | 0.929 | 1.198 | 0.901 | 1.293 | pr | 1 | **1.01E-05** | 1.21E-05 | 1.21E-05 | 1.21E-05 | 1.22E-05 |
| | 2 | **0.847** | 1.152 | 1.463 | 1.118 | 1.371 | | 2 | **9.96E-06** | 1.26E-05 | 1.26E-05 | 1.27E-05 | 1.26E-05 |
| | 3 | **0.878** | 1.293 | 1.650 | 1.263 | 1.418 | | 3 | **1.00E-05** | 1.28E-05 | 1.29E-05 | 1.29E-05 | 1.27E-05 |
| | 4 | **0.900** | 1.383 | 1.790 | 1.360 | 1.439 | | 4 | **1.01E-05** | 1.30E-05 | 1.31E-05 | 1.31E-05 | 1.29E-05 |
| | 5 | **0.920** | 1.447 | 1.899 | 1.436 | 1.448 | | 5 | **1.01E-05** | 1.32E-05 | 1.34E-05 | 1.33E-05 | 1.31E-05 |
| | 6 | **0.939** | 1.496 | 1.986 | 1.506 | 1.457 | | 6 | **1.01E-05** | 1.33E-05 | 1.36E-05 | 1.34E-05 | 1.31E-05 |
| | 7 | **0.958** | 1.529 | 2.054 | 1.558 | 1.469 | | 7 | **1.02E-05** | 1.34E-05 | 1.37E-05 | 1.36E-05 | 1.33E-05 |
| | 8 | **0.977** | 1.552 | 2.104 | 1.598 | 1.483 | | 8 | **1.03E-05** | 1.35E-05 | 1.39E-05 | 1.37E-05 | 1.33E-05 |
| | 9 | **0.984** | 1.570 | 2.140 | 1.612 | 1.498 | | 9 | **1.03E-05** | 1.36E-05 | 1.40E-05 | 1.38E-05 | 1.34E-05 |
| | 10 | **0.996** | 1.583 | 2.177 | 1.615 | 1.507 | | 10 | **1.04E-05** | 1.36E-05 | 1.41E-05 | 1.39E-05 | 1.35E-05 |
| | 11 | **0.999** | 1.596 | 2.226 | 1.620 | 1.507 | | 11 | **1.04E-05** | 1.37E-05 | 1.42E-05 | 1.40E-05 | 1.36E-05 |
| | 12 | **1.004** | 1.615 | 2.284 | 1.633 | 1.505 | | 12 | **1.04E-05** | 1.37E-05 | 1.43E-05 | 1.40E-05 | 1.36E-05 |
| psl | 1 | **154.569** | 192.223 | 192.649 | 189.818 | 191.848 | z500 | 1 | **19.833** | 24.322 | 24.825 | 24.160 | 24.566 |
| | 2 | **154.931** | 201.041 | 204.039 | 199.346 | 199.155 | | 2 | **19.926** | 25.503 | 26.731 | 25.988 | 25.328 |
| | 3 | **157.615** | 205.517 | 210.173 | 204.485 | 201.779 | | 3 | **20.377** | 26.042 | 27.453 | 27.019 | 25.655 |
| | 4 | **158.154** | 208.316 | 214.462 | 207.659 | 204.259 | | 4 | **20.646** | 26.359 | 27.906 | 27.630 | 25.941 |
| | 5 | **159.407** | 210.686 | 217.911 | 209.790 | 205.218 | | 5 | **21.007** | 26.713 | 28.175 | 28.275 | 26.113 |
| | 6 | **160.212** | 211.672 | 220.934 | 212.426 | 206.170 | | 6 | **21.305** | 26.960 | 28.580 | 29.056 | 26.205 |
| | 7 | **161.927** | 213.378 | 223.726 | 215.205 | 208.116 | | 7 | **21.748** | 27.283 | 29.026 | 29.632 | 26.484 |
| | 8 | **164.266** | 215.256 | 226.267 | 217.126 | 210.184 | | 8 | **22.209** | 27.514 | 29.468 | 30.007 | 26.844 |
| | 9 | **164.532** | 216.191 | 227.955 | 217.776 | 211.274 | | 9 | **22.499** | 27.733 | 29.770 | 30.190 | 27.163 |
| | 10 | **166.283** | 216.270 | 230.086 | 218.335 | 210.944 | | 10 | **22.712** | 27.818 | 30.237 | 30.287 | 27.377 |
| | 11 | **165.951** | 216.072 | 232.161 | 219.141 | 211.645 | | 11 | **22.756** | 27.840 | 30.860 | 30.343 | 27.565 |
| | 12 | **165.857** | 216.629 | 232.499 | 218.666 | 211.306 | | 12 | **22.776** | 27.929 | 31.294 | 30.365 | 27.588 |

Table 6: Global forecasting detail results of ClimateAR and baselines. The best results are bolded, and the second best results are underlined.

| var | lead | RMSE (↓) ClimateAR | Pangu | GraphCast | Oneforecast | ClimaX | DWD | ACC (↑) ClimateAR | Pangu | GraphCast | Oneforecast | ClimaX | DWD |
|---|---|---|---|---|---|---|---|---|---|---|---|---|---|
| t2m | 1 | 1.228 | 1.487 | 1.204 | **1.187** | 1.590 | 1.789 | **0.463** | 0.329 | 0.376 | 0.386 | 0.316 | 0.388 |
| | 2 | **1.349** | 1.777 | 1.462 | 1.457 | 1.685 | 1.871 | **0.334** | 0.208 | 0.230 | 0.234 | 0.234 | 0.319 |
| | 3 | **1.426** | 1.982 | 1.624 | 1.635 | 1.743 | 1.930 | 0.269 | 0.148 | 0.150 | 0.172 | 0.195 | **0.289** |
| | 4 | **1.490** | 2.137 | 1.726 | 1.747 | 1.767 | 1.978 | 0.237 | 0.107 | 0.097 | 0.132 | 0.172 | **0.271** |
| | 5 | **1.560** | 2.256 | 1.793 | 1.823 | 1.775 | 2.026 | 0.205 | 0.077 | 0.064 | 0.103 | 0.154 | **0.230** |
| | 6 | **1.622** | 2.344 | 1.842 | 1.891 | 1.784 | - | **0.189** | 0.056 | 0.035 | 0.075 | 0.135 | - |
| | 7 | **1.696** | 2.403 | 1.872 | 1.942 | 1.798 | - | **0.166** | 0.044 | 0.015 | 0.060 | 0.112 | - |
| | 8 | **1.771** | 2.440 | 1.894 | 1.985 | 1.817 | - | **0.141** | 0.031 | 0.001 | 0.037 | 0.094 | - |
| | 9 | **1.837** | 2.460 | 1.910 | 1.992 | 1.839 | - | **0.126** | 0.024 | -0.007 | 0.029 | 0.074 | - |
| | 10 | 1.913 | 2.488 | 1.918 | 1.990 | **1.849** | - | **0.107** | 0.018 | -0.012 | 0.014 | 0.067 | - |
| sst | 1 | 0.500 | 0.554 | 0.495 | **0.473** | 0.688 | 0.695 | 0.657 | **0.675** | 0.671 | 0.673 | 0.488 | 0.577 |
| | 2 | **0.625** | 0.823 | 0.693 | 0.654 | 0.763 | 0.803 | **0.476** | 0.425 | 0.427 | 0.437 | 0.330 | 0.469 |
| | 3 | **0.704** | 0.994 | 0.818 | 0.762 | 0.798 | 0.879 | 0.376 | 0.282 | 0.286 | 0.305 | 0.264 | **0.401** |
| | 4 | **0.771** | 1.117 | 0.902 | 0.833 | 0.819 | 0.938 | 0.315 | 0.195 | 0.194 | 0.224 | 0.232 | **0.351** |
| | 5 | **0.833** | 1.211 | 0.959 | 0.880 | **0.833** | 0.987 | 0.273 | 0.143 | 0.130 | 0.165 | 0.210 | **0.309** |
| | 6 | 0.898 | 1.292 | 1.003 | 0.919 | **0.848** | - | **0.242** | 0.102 | 0.083 | 0.122 | 0.187 | - |
| | 7 | 0.968 | 1.365 | 1.038 | 0.958 | **0.867** | - | **0.215** | 0.071 | 0.054 | 0.088 | 0.165 | - |
| | 8 | 1.039 | 1.424 | 1.067 | 0.990 | **0.884** | - | **0.194** | 0.047 | 0.034 | 0.068 | 0.141 | - |
| | 9 | 1.111 | 1.468 | 1.094 | 1.016 | **0.900** | - | **0.172** | 0.031 | 0.018 | 0.048 | 0.110 | - |
| | 10 | 1.182 | 1.505 | 1.116 | 1.033 | **0.916** | - | **0.153** | 0.022 | 0.009 | 0.031 | 0.088 | - |
| t500 | 1 | **1.069** | 1.154 | 1.164 | 1.158 | 1.160 | - | **0.363** | 0.304 | 0.248 | 0.268 | 0.320 | - |
| | 2 | **1.107** | 1.233 | 1.239 | 1.256 | 1.194 | - | **0.305** | 0.245 | 0.159 | 0.184 | 0.279 | - |
| | 3 | **1.133** | 1.293 | 1.274 | 1.316 | 1.220 | - | **0.282** | 0.217 | 0.115 | 0.159 | 0.264 | - |
| | 4 | **1.162** | 1.341 | 1.291 | 1.351 | 1.243 | - | **0.265** | 0.192 | 0.080 | 0.130 | 0.239 | - |
| | 5 | **1.194** | 1.385 | 1.304 | 1.378 | 1.252 | - | **0.248** | 0.177 | 0.056 | 0.107 | 0.227 | - |
| | 6 | **1.228** | 1.423 | 1.320 | 1.422 | 1.254 | - | **0.236** | 0.158 | 0.032 | 0.092 | 0.224 | - |
| | 7 | **1.264** | 1.460 | 1.335 | 1.453 | 1.267 | - | **0.212** | 0.140 | 0.001 | 0.065 | 0.202 | - |
| | 8 | 1.304 | 1.489 | 1.343 | 1.467 | **1.284** | - | **0.177** | 0.114 | -0.010 | 0.046 | 0.174 | - |
| | 9 | 1.336 | 1.513 | 1.353 | 1.472 | **1.299** | - | **0.151** | 0.098 | -0.025 | 0.030 | 0.145 | - |
| | 10 | 1.387 | 1.552 | 1.355 | 1.480 | **1.310** | - | 0.118 | 0.070 | -0.030 | 0.006 | **0.123** | - |
| t850 | 1 | **1.235** | 1.500 | 1.352 | 1.338 | 1.424 | - | **0.324** | 0.244 | 0.229 | 0.242 | 0.264 | - |
| | 2 | **1.288** | 1.673 | 1.468 | 1.477 | 1.481 | - | **0.246** | 0.172 | 0.136 | 0.143 | 0.219 | - |
| | 3 | **1.319** | 1.787 | 1.533 | 1.562 | 1.518 | - | **0.219** | 0.134 | 0.083 | 0.110 | 0.202 | - |
| | 4 | **1.350** | 1.858 | 1.569 | 1.608 | 1.544 | - | **0.200** | 0.111 | 0.052 | 0.083 | 0.171 | - |
| | 5 | **1.385** | 1.906 | 1.592 | 1.641 | 1.550 | - | **0.176** | 0.094 | 0.030 | 0.061 | 0.161 | - |
| | 6 | **1.419** | 1.952 | 1.614 | 1.686 | 1.555 | - | **0.165** | 0.079 | 0.014 | 0.042 | 0.150 | - |
| | 7 | **1.465** | 2.003 | 1.631 | 1.715 | 1.567 | - | **0.143** | 0.058 | -0.003 | 0.028 | 0.127 | - |
| | 8 | **1.511** | 2.042 | 1.645 | 1.735 | 1.583 | - | **0.121** | 0.044 | -0.009 | 0.013 | 0.103 | - |
| | 9 | **1.548** | 2.083 | 1.657 | 1.738 | 1.599 | - | **0.110** | 0.031 | -0.014 | 0.007 | 0.084 | - |
| | 10 | **1.600** | 2.142 | 1.660 | 1.736 | 1.608 | - | **0.089** | 0.023 | -0.016 | -0.002 | 0.077 | - |
| t1000 | 1 | **1.194** | 1.455 | 1.358 | 1.390 | 1.756 | - | **0.444** | 0.333 | 0.310 | 0.324 | 0.271 | - |
| | 2 | **1.308** | 1.720 | 1.534 | 1.608 | 1.789 | - | **0.321** | 0.207 | 0.194 | 0.199 | 0.194 | - |
| | 3 | **1.379** | 1.878 | 1.630 | 1.731 | 1.820 | - | **0.258** | 0.142 | 0.123 | 0.146 | 0.155 | - |
| | 4 | **1.438** | 1.973 | 1.684 | 1.800 | 1.842 | - | **0.226** | 0.098 | 0.077 | 0.106 | 0.129 | - |
| | 5 | **1.501** | 2.031 | 1.721 | 1.841 | 1.856 | - | **0.194** | 0.070 | 0.050 | 0.080 | 0.112 | - |
| | 6 | **1.559** | 2.070 | 1.750 | 1.882 | 1.863 | - | **0.179** | 0.047 | 0.027 | 0.054 | 0.098 | - |
| | 7 | **1.627** | 2.100 | 1.766 | 1.916 | 1.871 | - | **0.158** | 0.033 | 0.012 | 0.035 | 0.080 | - |
| | 8 | **1.697** | 2.121 | 1.778 | 1.944 | 1.882 | - | **0.135** | 0.023 | 0.003 | 0.015 | 0.062 | - |
| | 9 | **1.759** | 2.140 | 1.791 | 1.941 | 1.891 | - | **0.124** | 0.013 | 0.000 | 0.015 | 0.048 | - |
| | 10 | 1.834 | 2.168 | **1.798** | 1.938 | 1.899 | - | **0.105** | 0.015 | -0.002 | 0.003 | 0.045 | - |
| z500 | 1 | **30.096** | 32.268 | 31.120 | 31.201 | 32.974 | 38.268 | **0.404** | 0.327 | 0.282 | 0.311 | 0.367 | 0.394 |
| | 2 | **31.209** | 34.459 | 32.624 | 33.584 | 34.104 | 41.048 | 0.354 | 0.270 | 0.176 | 0.201 | 0.315 | **0.360** |
| | 3 | **31.956** | 35.288 | 33.206 | 34.789 | 34.498 | 42.861 | 0.330 | 0.258 | 0.131 | 0.167 | 0.300 | **0.344** |
| | 4 | **32.661** | 35.886 | 33.592 | 35.531 | 34.841 | 44.424 | 0.312 | 0.231 | 0.091 | 0.139 | 0.276 | **0.344** |
| | 5 | **33.372** | 36.229 | 34.053 | 36.249 | 35.038 | 46.368 | 0.294 | 0.216 | 0.049 | 0.108 | 0.252 | **0.298** |
| | 6 | **34.033** | 36.584 | 34.297 | 37.064 | 35.172 | - | **0.278** | 0.194 | 0.013 | 0.083 | 0.232 | - |
| | 7 | 34.935 | 36.871 | 34.674 | 37.643 | 35.609 | - | **0.243** | 0.175 | -0.023 | 0.064 | 0.214 | - |
| | 8 | 35.895 | 37.095 | 34.962 | 38.062 | 36.429 | - | **0.214** | 0.139 | -0.039 | 0.052 | 0.191 | - |
| | 9 | 36.937 | 37.189 | 35.200 | 38.288 | 36.767 | - | **0.173** | 0.102 | -0.053 | 0.036 | 0.139 | - |
| | 10 | 38.270 | 37.485 | **35.244** | 38.458 | 36.701 | - | **0.137** | 0.049 | -0.059 | 0.006 | 0.131 | - |
| z850 | 1 | **19.573** | 20.017 | 20.271 | 20.052 | 19.881 | 22.619 | 0.334 | 0.291 | 0.210 | 0.261 | **0.337** | 0.309 |
| | 2 | **20.009** | 20.961 | 21.053 | 20.978 | 20.626 | 23.007 | **0.272** | 0.216 | 0.125 | 0.170 | 0.259 | 0.262 |
| | 3 | **20.315** | 21.443 | 21.316 | 21.353 | 20.814 | 23.292 | 0.253 | 0.190 | 0.087 | 0.137 | 0.251 | **0.256** |
| | 4 | **20.540** | 21.867 | 21.497 | 21.579 | 21.039 | 23.462 | 0.245 | 0.139 | 0.057 | 0.104 | 0.212 | **0.258** |
| | 5 | **20.851** | 22.093 | 21.702 | 21.784 | 21.150 | 24.194 | 0.233 | 0.108 | 0.021 | 0.076 | 0.191 | **0.204** |
| | 6 | **21.021** | 22.258 | 21.698 | 22.015 | 21.220 | - | **0.236** | 0.085 | 0.008 | 0.054 | 0.184 | - |
| | 7 | 21.431 | 22.442 | 21.835 | 22.225 | **21.407** | - | **0.207** | 0.068 | -0.005 | 0.038 | 0.169 | - |

Continued on next page

Continued

| Var | | | | | | | | | | | | | |
|---|---|---|---|---|---|---|---|---|---|---|---|---|---|
| | 8 | 21.872 | 22.662 | 21.983 | 22.395 | **21.625** | - | **0.198** | 0.051 | -0.022 | 0.041 | 0.138 | - |
| | 9 | 22.445 | 22.885 | 22.046 | 22.520 | **21.758** | - | **0.180** | 0.026 | -0.029 | 0.034 | 0.109 | - |
| | 10 | 23.183 | 23.168 | 22.044 | 22.614 | **21.789** | - | **0.152** | -0.001 | -0.036 | 0.011 | 0.106 | - |
| z1000 | 1 | **19.232** | 19.839 | 20.222 | 20.125 | 21.193 | 22.719 | 0.348 | 0.334 | 0.245 | 0.293 | **0.355** | 0.314 |
| | 2 | **19.703** | 21.033 | 21.251 | 21.326 | 22.161 | 22.891 | **0.272** | 0.232 | 0.145 | 0.195 | 0.263 | 0.260 |
| | 3 | **20.054** | 21.594 | 21.692 | 21.811 | 22.378 | 22.960 | 0.243 | 0.207 | 0.099 | 0.169 | **0.245** | 0.244 |
| | 4 | **20.331** | 22.029 | 21.979 | 22.094 | 22.629 | 22.963 | **0.232** | 0.162 | 0.065 | 0.130 | 0.201 | **0.247** |
| | 5 | **20.712** | 22.256 | 22.220 | 22.296 | 22.711 | 23.549 | **0.213** | 0.132 | 0.036 | 0.097 | 0.170 | 0.186 |
| | 6 | **20.916** | 22.422 | 22.262 | 22.505 | 22.751 | - | **0.216** | 0.105 | 0.026 | 0.071 | 0.157 | - |
| | 7 | **21.333** | 22.589 | 22.407 | 22.725 | 22.959 | - | **0.189** | 0.083 | 0.022 | 0.049 | 0.138 | - |
| | 8 | **21.764** | 22.734 | 22.588 | 22.903 | 23.227 | - | **0.185** | 0.062 | 0.005 | 0.041 | 0.108 | - |
| | 9 | **22.289** | 22.899 | 22.674 | 23.021 | 23.379 | - | **0.180** | 0.034 | -0.005 | 0.035 | 0.082 | - |
| | 10 | 22.999 | 23.123 | 22.664 | 23.123 | 23.362 | - | **0.157** | 0.017 | -0.005 | 0.023 | 0.087 | - |
| pr($\times 10^{-5}$) | 1 | **1.648** | 1.682 | 1.675 | 1.678 | 1.702 | 2.080 | 0.181 | 0.163 | 0.146 | 0.161 | **0.185** | 0.138 |
| | 2 | **1.667** | 1.735 | 1.725 | 1.733 | 1.742 | 2.097 | **0.139** | 0.126 | 0.106 | 0.121 | 0.133 | 0.104 |
| | 3 | **1.684** | 1.772 | 1.759 | 1.770 | 1.765 | 2.119 | 0.117 | 0.107 | 0.080 | 0.103 | **0.117** | 0.097 |
| | 4 | **1.705** | 1.803 | 1.784 | 1.796 | 1.786 | 2.151 | **0.103** | 0.087 | 0.059 | 0.081 | 0.095 | 0.085 |
| | 5 | **1.726** | 1.833 | 1.803 | 1.817 | 1.804 | 2.178 | **0.093** | 0.067 | 0.046 | 0.063 | 0.086 | 0.071 |
| | 6 | **1.746** | 1.855 | 1.816 | 1.834 | 1.814 | - | **0.091** | 0.054 | 0.036 | 0.052 | 0.077 | - |
| | 7 | **1.777** | 1.877 | 1.829 | 1.854 | 1.829 | - | **0.080** | 0.038 | 0.028 | 0.045 | 0.062 | - |
| | 8 | **1.812** | 1.893 | 1.842 | 1.872 | 1.840 | - | **0.066** | 0.018 | 0.017 | 0.028 | 0.047 | - |
| | 9 | **1.854** | 1.904 | 1.853 | 1.886 | 1.851 | - | **0.061** | 0.007 | 0.010 | 0.019 | 0.034 | - |
| | 10 | 1.908 | 1.912 | 1.863 | 1.899 | **1.860** | - | **0.056** | -0.003 | 0.004 | 0.009 | 0.028 | - |
| psl | 1 | **237.243** | 245.675 | 245.801 | 242.856 | 243.295 | 284.758 | 0.351 | 0.309 | 0.249 | 0.301 | **0.360** | 0.315 |
| | 2 | **243.172** | 259.654 | 256.854 | 255.864 | 253.370 | 289.515 | **0.274** | 0.212 | 0.151 | 0.199 | 0.266 | 0.259 |
| | 3 | **247.742** | 267.181 | 261.879 | 262.011 | 256.714 | 290.805 | 0.245 | 0.182 | 0.103 | 0.172 | **0.246** | 0.244 |
| | 4 | **251.438** | 272.706 | 265.405 | 265.906 | 259.874 | 291.080 | 0.233 | 0.148 | 0.067 | 0.133 | 0.201 | **0.247** |
| | 5 | **256.362** | 276.614 | 268.287 | 268.545 | 261.215 | 298.682 | **0.213** | 0.123 | 0.041 | 0.101 | 0.171 | 0.185 |
| | 6 | **259.028** | 279.888 | 269.181 | 271.727 | 262.304 | - | **0.215** | 0.095 | 0.030 | 0.076 | 0.159 | - |
| | 7 | **264.323** | 282.558 | 271.174 | 274.927 | 264.772 | - | **0.189** | 0.079 | 0.024 | 0.052 | 0.140 | - |
| | 8 | **269.725** | 284.993 | 273.249 | 277.126 | 267.130 | - | **0.185** | 0.051 | 0.007 | 0.043 | 0.110 | - |
| | 9 | 276.206 | 286.517 | 274.057 | 278.237 | **268.441** | - | **0.180** | 0.029 | -0.001 | 0.036 | 0.086 | - |
| | 10 | 284.973 | 288.660 | 273.684 | 279.112 | **268.147** | - | **0.158** | 0.008 | 0.000 | 0.027 | 0.088 | - |
| v1000 | 1 | **1.099** | 1.175 | 1.145 | 1.150 | 1.224 | - | 0.217 | 0.197 | 0.166 | 0.194 | **0.218** | - |
| | 2 | **1.112** | 1.242 | 1.206 | 1.220 | 1.260 | - | **0.167** | 0.136 | 0.110 | 0.134 | 0.157 | - |
| | 3 | **1.124** | 1.285 | 1.241 | 1.260 | 1.274 | - | 0.142 | 0.110 | 0.081 | 0.117 | **0.145** | - |
| | 4 | **1.139** | 1.314 | 1.265 | 1.283 | 1.285 | - | **0.122** | 0.086 | 0.059 | 0.098 | 0.121 | - |
| | 5 | **1.150** | 1.333 | 1.278 | 1.296 | 1.292 | - | **0.112** | 0.074 | 0.042 | 0.078 | 0.109 | - |
| | 6 | **1.161** | 1.348 | 1.286 | 1.310 | 1.299 | - | **0.112** | 0.059 | 0.030 | 0.057 | 0.091 | - |
| | 7 | **1.184** | 1.360 | 1.293 | 1.322 | 1.309 | - | **0.094** | 0.052 | 0.030 | 0.046 | 0.071 | - |
| | 8 | **1.205** | 1.370 | 1.300 | 1.331 | 1.319 | - | **0.083** | 0.035 | 0.013 | 0.032 | 0.051 | - |
| | 9 | **1.231** | 1.380 | 1.306 | 1.340 | 1.326 | - | **0.070** | 0.010 | 0.003 | 0.019 | 0.036 | - |
| | 10 | **1.258** | 1.390 | 1.310 | 1.346 | 1.329 | - | **0.063** | 0.003 | 0.002 | 0.003 | 0.031 | - |
| u1000 | 1 | **1.381** | 1.464 | 1.447 | 1.443 | 1.525 | - | 0.226 | 0.205 | 0.166 | 0.203 | **0.233** | - |
| | 2 | **1.409** | 1.554 | 1.540 | 1.546 | 1.569 | - | 0.170 | 0.142 | 0.096 | 0.134 | **0.170** | - |
| | 3 | **1.424** | 1.604 | 1.588 | 1.603 | 1.585 | - | 0.151 | 0.120 | 0.066 | 0.110 | **0.152** | - |
| | 4 | **1.444** | 1.640 | 1.616 | 1.638 | 1.601 | - | 0.139 | 0.096 | 0.050 | 0.089 | **0.120** | - |
| | 5 | **1.460** | 1.665 | 1.638 | 1.661 | 1.611 | - | 0.132 | 0.080 | 0.035 | 0.070 | 0.107 | - |
| | 6 | **1.478** | 1.684 | 1.649 | 1.678 | 1.617 | - | **0.130** | 0.064 | 0.022 | 0.056 | 0.099 | - |
| | 7 | **1.506** | 1.700 | 1.663 | 1.700 | 1.628 | - | **0.114** | 0.054 | 0.015 | 0.039 | 0.090 | - |
| | 8 | **1.538** | 1.715 | 1.676 | 1.715 | 1.642 | - | **0.099** | 0.033 | 0.004 | 0.031 | 0.068 | - |
| | 9 | **1.572** | 1.726 | 1.685 | 1.731 | 1.654 | - | **0.094** | 0.011 | -0.004 | 0.012 | 0.045 | - |
| | 10 | **1.615** | 1.738 | 1.688 | 1.739 | 1.656 | - | **0.083** | -0.004 | -0.008 | 0.001 | 0.038 | - |

**Region Forecasting.** In Fig. 9, we give the comparison of ACC for ClimateAR and baselines in the low latitude regional forecast (i.e. 30°S to 30°N.)across 1- to 14-month lead times. The full results demonstrate that ClimateAR can significantly extend its effective forecast step over yearly in regions with greater predictability (i.e. low latitued region).

## A.4.2 VISUALIZATION

To further demonstrate ClimateAR's forecasting capability across different regions, we visualize the RMSE distribution of Sea Surface Temperature (sst) and the ACC distribution of other variables in Fig. 10, 11, 13, 12. From the extensive results, we observe that ClimateAR performs particularly well in regions with high predictability (e.g., the tropics and the eastern Pacific—on seasonal scales). This highlights the model's ability to capture climate modes critical for long-term climate prediction.

## A.4.3 POWER SPECTRAL DENSITY ANALYZE

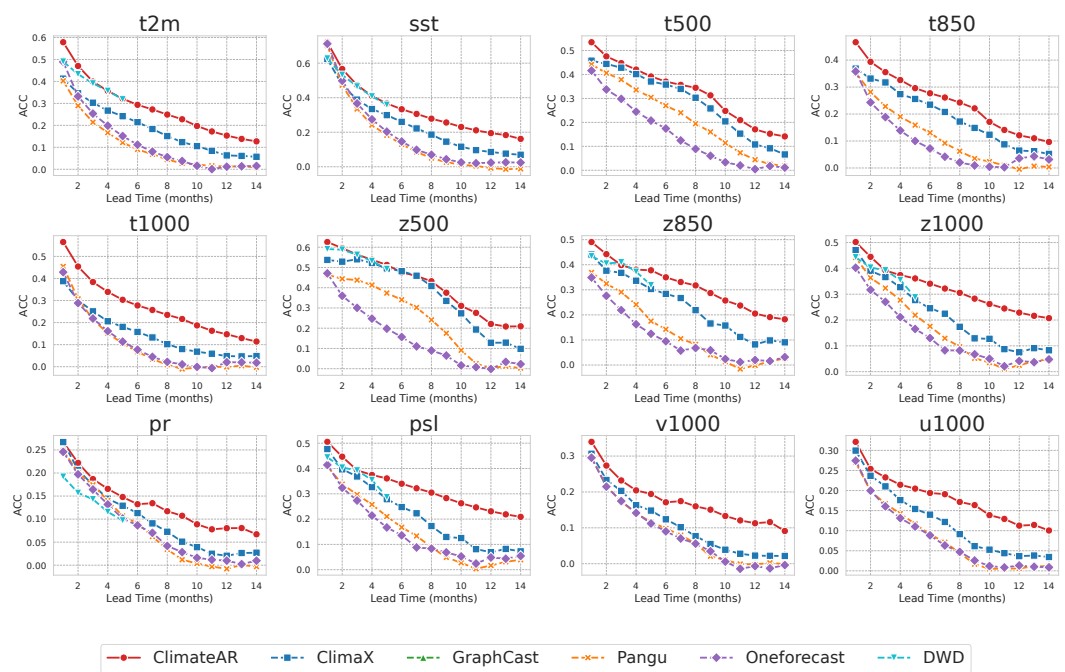

Figure 9: Comparison of ACC for ClimateAR and baselines in the low latitude regional forecast across 1- to 14-month lead times.

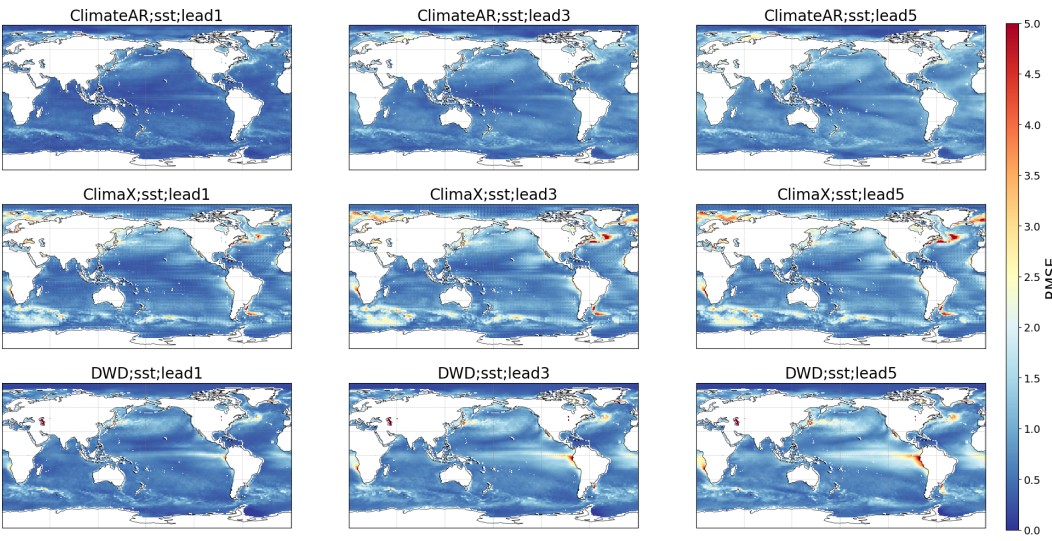

Figure 10: The global RMSE distribution of sea surface temperature forecasting with 1-, 3-, and 5-month lead times in the testing set.

We consider the power spectral density (Hickman et al., 2025) of the Niño3.4 index and Indian Ocean Dipole (IOD) index in Fig. 14. We can find that ClimateAR successfully captures the primary peaks of different climate indices at interannual timescale.

### A.4.4 MULTI-SCALE PROCESSES INTERACTIONS

**Global and Regional Scale.** To evaluate the ability of ClimateAR to capture interactions across multiple scales, we examine the relationship between ENSO and global mean surface temperature

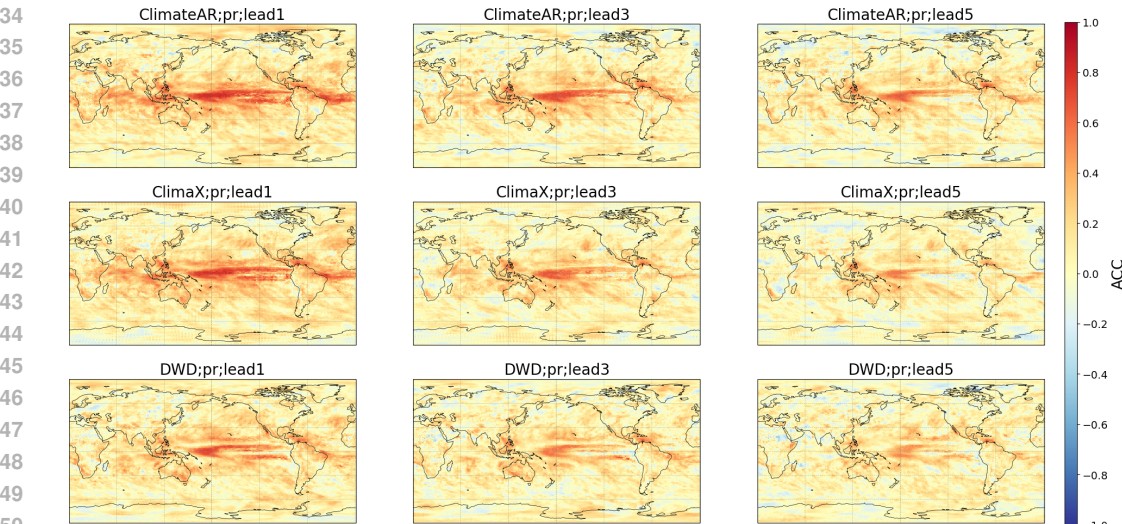

Figure 11: The global ACC distribution of precipitation rate forecasting with 1-, 3-, and 5-month lead times in the testing set.

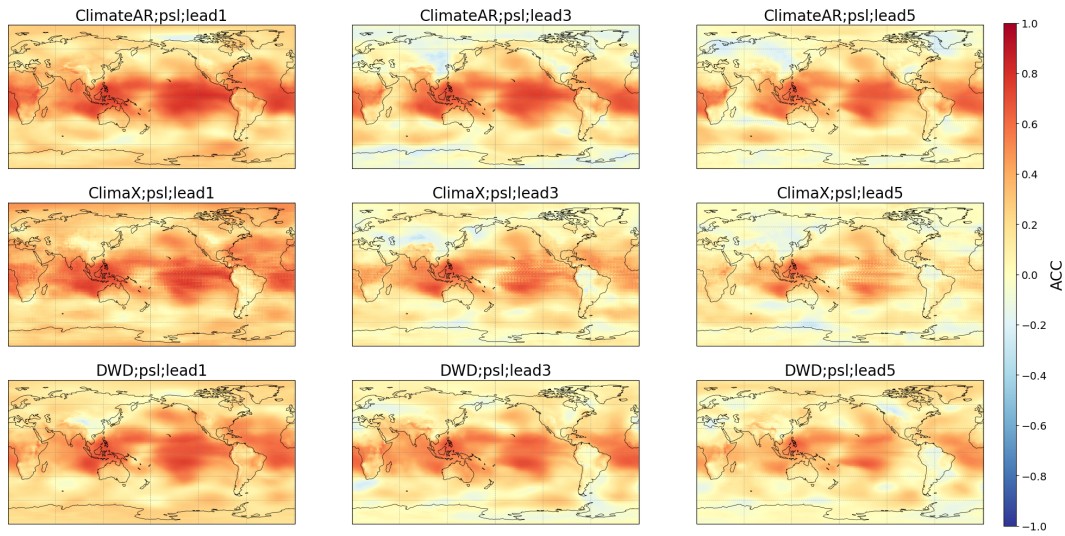

Figure 12: The global ACC distribution of sea level pressure forecasting with 1-, 3-, and 5-month lead times in the testing set.

(GMST) anomaly. In Fig. 15, we present the time series of the Niño3.4 index and GMST anomaly from both observations and predictions, together with their corresponding regression coefficients. The regression coefficients between ENSO and GMST in the predictions is nearly identical to those in the ERA5 dataset (difference less than $0.05K$), and the predicted values accurately reproduce the amplitude and phase of the observed variations. This shows that the model effectively captures the short-term temperature fluctuations associated with ENSO events and successfully reproduces the expected lagged response of GMST to ENSO forcing (typically about 3-6 months).

**Regional and Local Scale.** Fig. 16 shows the spatial distribution of regression coefficients between ENSO and pr/t2m anomaly at each grid. The predicted patterns closely resemble those observed. The model accurately identifies the characteristic teleconnections associated with ENSO and captures the positive or negative grid-point-level responses to ENSO intensity across the global domain.

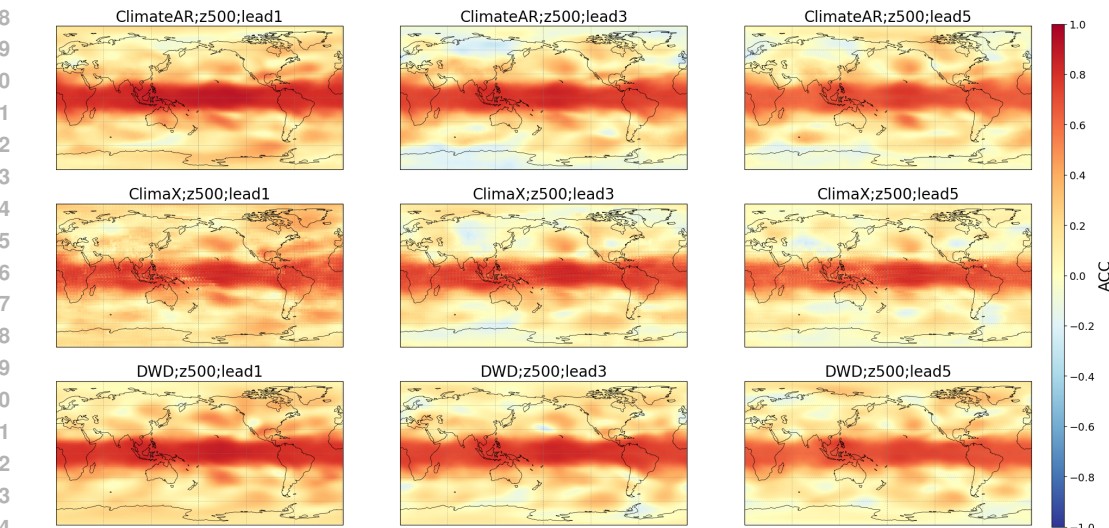

Figure 13: The global ACC distribution of 500hPa geopotential forecasting with 1-, 3-, and 5-month lead times in the testing set.

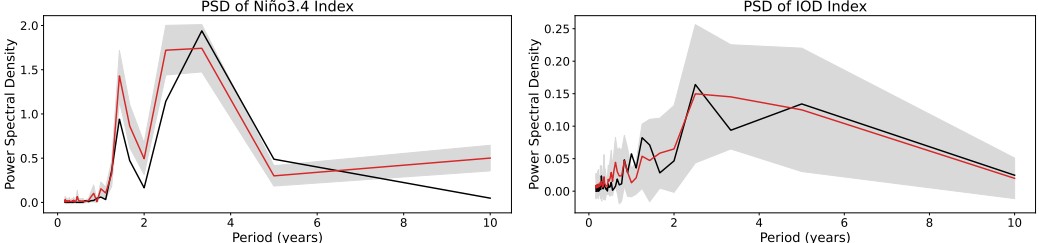

Figure 14: The Power Spectral Density (PSD) of the Niño3.4 index generated by CliamteAR at 6-month lead time and IOD index generated by CliamteAR at 1-month lead time and compare it with the reference data. The shaded area represents the range indicating twice the ensemble standard deviation.

Overall, GMST represents large-scale global variability, ENSO indices represents mid-scale variability, and grid-wise precipitation/t2m anomalies represents small-scale regional fluctuations. The high consistency between predicted and observed relationships across all these scales indicates that ClimateAR effectively models the interactions among climate processes and successfully captures cross-scale variability.

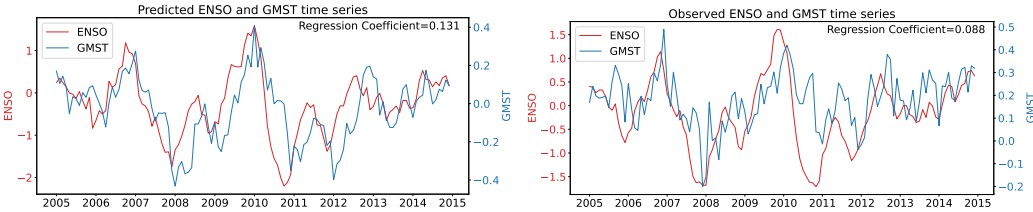

Figure 15: Comparison of the prediction ahead 1 month and observation for the global mean surface temperature (GMST) and ENSO time series.

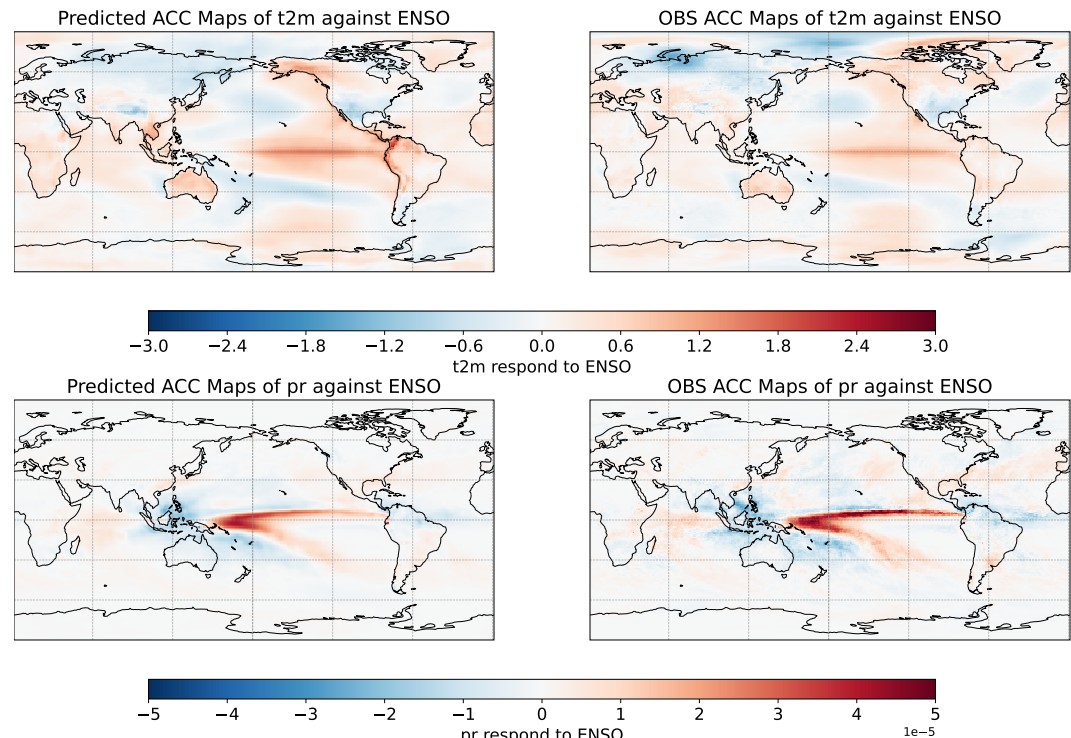

Figure 16: Maps of regression coefficients of predicted ahead 1 month and observed pr and t2m respond to the Niño3.4 index over 2005–2014.

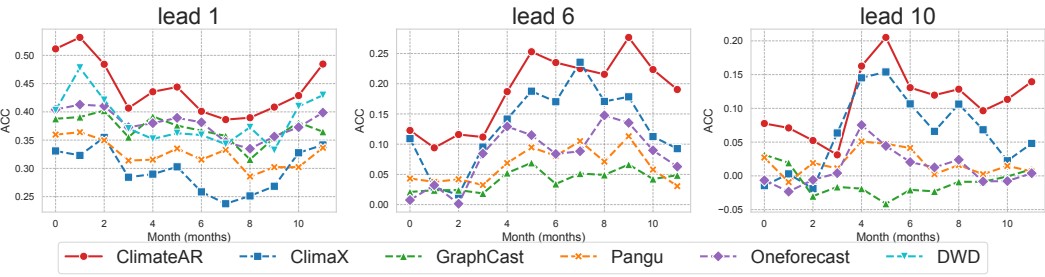

Figure 17: T2m ACC forecast starting from each month at 1-,6-,and 10-lead times.

### A.4.5 FORECASTING FROM DIFFERENT MONTHS

In Fig. 17, we use each of the 12 months in a year as a forecasting start point with lead times of 1, 6, and 10 months, respectively, and compute the average t2m forecast ACC values. It can be seen that ClimateAR achieved the best ACC level in most months. For some months that are difficult to predict (e.g., the months when the ENSO starting phase is reset), we can also get high ACC performance.

### A.4.6 HYPERPARAMETER STUDY

We conducted a search for two critical hyperparameters during the model's pre-training phase, including noise ratio $p$ and number of partitions $N$. Specifically, $p$ controls the noise ratio of the teach-forcing input during the training procedure, $N$ controls the number of partitions codebooks and $V$ controls the size of codebooks. To ensure fairness, other hyperparameters are fixed when a specific one is tuned. The details of different ACCs for different values of $p$ and $N$ are summarized in Table 7.

Table 7: 1 month-ahead prediction ACC for different hyperparameters of ClimateAR.

| | ACC (↑) | | | | | | | | | | | |
|---|---|---|---|---|---|---|---|---|---|---|---|---|
| | noise | | | | codebooks | | | | codebook size | | | |
| | 0.2 | 0.3 | 0.4 | 0.5 | 2 | 4 | 8 | 16 | 1024 | 2048 | 4096 | 8192 |
| z500 | 0.408 | **0.428** | 0.414 | 0.405 | 0.395 | 0.394 | **0.428** | 0.398 | 0.402 | 0.408 | **0.428** | 0.397 |
| t2m | 0.456 | **0.480** | 0.462 | 0.466 | 0.445 | 0.455 | **0.480** | 0.467 | 0.463 | 0.467 | **0.480** | 0.466 |
| pr | 0.162 | **0.190** | 0.171 | 0.164 | 0.159 | 0.162 | **0.190** | 0.164 | 0.164 | 0.162 | **0.190** | 0.161 |
| psl | 0.348 | **0.374** | 0.356 | 0.344 | 0.327 | 0.331 | **0.374** | 0.335 | 0.350 | 0.350 | **0.374** | 0.330 |

Table 8: Result of efficiency study of ClimateAR and data-driven baselines.

| Method | Parameters | GPU Memory (GB) | FLOPs | Training Time Min / Epoch | Inference Time s / Step | ACC |
|---|---|---|---|---|---|---|
| ClimaX | 104.0 M | 19.27 | 420.97 G | 18.78 | 0.162 | 0.376 |
| Oneforecast | 24.7 M | 38.95 | 548.78 G | 69.73 | 0.516 | 0.386 |
| GraphCast | 28.9 M | 34.59 | 1852.26 G | 32.33 | 0.279 | 0.376 |
| Pangu | 23.9M | 7.60 | 213.82G | 5.48 | 0.047 | 0.329 |
| ClimateAR | 480.6 M | 18.34 | 324.23 G | 13.13 | 0.179 | 0.464 |

### A.4.7 EFFICIENCY STUDY

**Training Cost.** The training time complexities of ClimateAR are $O(h^2w^2)$, where $(h, w)$ is the size of feature map $\mathbf{f}$. which is consistent with traditional ViTs. To conduct a fair comparison, we uniformly set the batch size to 8 when measuring the GPU memory usage.

**Inference Cost.** Unlike traditional autoregressive models, the inference time complexities of ClimateAR are consistent with the complexity of the training $O(h^2w^2)$, which has been demonstrated in previous paper (Tian et al., 2024). When the number of predicted members in the set is $M$, the complexities increases to $O(Mh^2w^2)$.

From Table 8 we can find that (1) As a generative model, ClimateAR has a lower cost than most deterministic models, which demonstrates a competitive efficiency. (2) Due to the patch-based processing, the training time of ClimaX, Pangu, and ClimateAR is significantly lower than that of deterministic graph-based models.

### A.5 ROLLING FINE-TUNE

On ERA5 dataset, we use the model to generate predictions for 1 to 3 steps in a rolling fashion and then mix these predictions into our sample input. This allows the model to get the ability to correct accumulated errors as it optimizes the loss function with a biased input. After rolling fine-tuning procedure, we successfully mitigated the RMSE drift of the model, and some of the results are summarized in Fig. 18.

### A.6 ADDITIONAL RELATE WORK

**AutoRegressive Models.** AutoRegressive models have achieved significant success in language and visual generation tasks (Radford et al., 2019; Bai et al., 2024). Early visual AR models generate images at pixel level row-by-row (Van den Oord et al., 2016; Van Den Oord et al., 2016; Chen et al., 2020), whereas later models inspired by ViT (Dosovitskiy et al., 2021) generate images at patch level row-by-row (Esser et al., 2021; Lee et al., 2022). Recent advances in visual AR models based on next-scale prediction (Tian et al., 2024) have significantly improved generative quality and speed, and demonstrated scaling capabilities comparable to large language models (LLMs) (Kaplan et al., 2020; Hoffmann et al., 2022). Compared to diffusion models, AR models exhibit superior efficiency in both training and inference, and are particularly advantageous for fusing complex multimodal conditional information with discrete tokens (Yu et al., 2023; Lu et al., 2023). While diffusion-based generative models (Price et al., 2025; Oskarsson et al., 2024) and deterministic models(Nguyen et al., 2023; Bi et al., 2023) have been extensively explored for weather and climate forecasting, the potential of AR generative models in long-term climate forecasting remains underexplored.

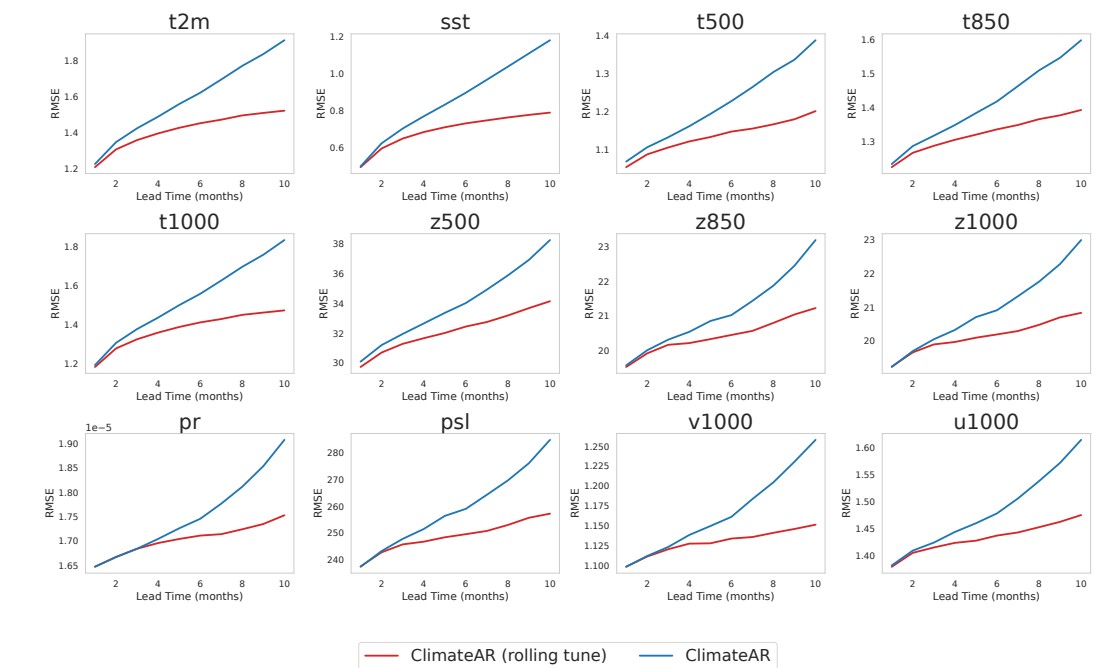

Figure 18: Comparison of global forecast RMSE for ClimateAR before and after rolling fine-tuning across 1- to 10-month lead times.

## A.7 LIMITATION AND FUTURE WORK

Despite such advantages, there are several directions that remain to be explored in the future: (1) fine-tuning the model to reduce the cumulative loss in iterative forecasts; (2) incorporating contrastive learning or classifier-free guidance techniques in generative models to enhance the model's ability to predict anomalies; and (3) using ClimateAR to assist in more tasks than forecasting such as dimensionality reduction analysis and anomaly detection.

