# OpenReview forum: "ClimateAR: Multi-Scale Autoregressive Generative Modeling for Climate Forecasting"
_ICLR.cc/2026/Conference — Submitted to ICLR 2026_

### Official Review · Reviewer_36PM · 2025-10-22

**Soundness:** 3
**Presentation:** 3
**Contribution:** 2
**Rating:** 4
**Confidence:** 4

**Summary:**

The paper introduces ClimateAR, an autoregressive generative model for probabilistic climate forecasting. It addresses limitations in existing deterministic and generative models by capturing multi-scale spatiotemporal dependencies and climate uncertainty. ClimateAR features two key innovations: an aligned tokenizer for bridging simulation and real-world data, and a mixed-scale conditioning mechanism to model cross-scale climate interactions. Extensive evaluations and training is done on the range of datasets including ERA5, CMIP6 and ORAS5 datasets. The results show that ClimateAR outperforms state-of-the-art weather forecasting models. The model demonstrates strong performance in forecasting critical climate phenomena like El Niño–Southern Oscillation (ENSO). ClimateAR is computationally efficient and offers robust transferability across simulated and real-world datasets.

**Strengths:**

The paper addresses critical gaps in existing models by incorporating multi-scale spatiotemporal dependencies and explicitly modeling climate uncertainty. The use of an aligned tokenizer to bridge simulated and real-world data and a mixed-scale conditioning mechanism to capture cross-scale interactions represents a creative combination of existing ideas tailored to the unique challenges of climate forecasting. The model's robustness is validated through zero-shot forecasting, ENSO prediction, and ablation studies, showcasing its adaptability and effectiveness.

**Weaknesses:**

One of paper's main claim is "the capture of inherent climate uncertainty". However, only RMSE and ACC are used as evaluation metrics and it is nowhere quantified in the paper of how uncertain or certain the model is. If authors can run experiments on few ensembles and quantify uncertainty using metrics like CRPS or log likelihood. This would strengthen the paper.
When referred to Climate Forecasting, it goes beyond monthly forecasts upto several years or decadal forecasts. The results are shown upto 10 months only which in my opinion doesn't justify the climate forecasting part.
The evaluation done against the data-driven models such as graphcast, pangu weather were inherently built for weather forecasting (from short-term to medium-range forecasting upto 10-15 days, not climate forecasting.
Also, the RMSE results show the performance tend to de grade after 8 months lead time on several variables.

**Questions:**

Please address some of the concerns mentioned in weakness sections.
I think the problem needs to be reframed as long-term weather forecasting instead of climate forecasting.
OR
The forecasting and evaluation framework needs to be adjusted according to the climate forecast problem.
There are many works on climate forecasting that goes beyond yearly forecasts. Please have a look at them. Few of those works include ACE2 and NeuralGCM.
If the framework and evaluation is reformulated fairly, I'll consider changing my score.

---

> ### Author Response · Authors · 2025-11-22
> **Response to Reviewer 36PM (Part 1)**
>
> We sincerely thank you for your constructive comments of our work. We have carefully considered your concerns regarding probabilistic forecasting and climate forecasting. Below, we provide detailed responses and new experimental evidence to address these points.
>
> >Weakness 1 and question 1: Probabilistic measurements such as CRPS or log likelihood, or even a few ensembles, and quantify uncertainty.
>
> Thank you for suggesting adding probabilistic forecasting experiments. We have provided two important experiments to demonstrate this:
>
> (a) We evaluated the **continuous ranked probability score (CRPS)** to quantify uncertainty. For the deterministic baseline, we obtained different ensemble forecast members through slight perturbations in the initial field. The results of CRPS are as follows:
>
> |CRPS|lead	|ClimateAR	|GraphCast|	Pangu	|Oneforecast	|ClimaX	|CRPS |lead	|ClimateAR	|GraphCast|	Pangu	|Oneforecast	|ClimaX	|
> | :---:  |  :---:  | :---:  |  :---:  | :---:  |  :---:  | :---:  | :---:  |  :---:  | :---:  |  :---:  | :---: | :---: | :---: |
> |t2m|	1	|**0.831** |0.929 |1.198 |0.901 |1.293 |pr $(\times 10^{-5})$| 1|**1.005**|1.205|	1.209|1.211|1.222|
> |   | 2	|**0.847**   |1.152 |1.463 |1.118 |1.371 |	| 2	|**1.000**	|1.256	|1.258	|1.265	|1.256|
> |   | 3	|**0.878**   |1.293 |1.650 |1.263 |1.418 |	| 3	|**1.002**	|1.285	|1.290	|1.295	|1.275|
> |   | 4	|**0.900**   |1.383 |1.790 |1.360 |1.439 |	| 4	|**1.008**	|1.304	|1.314	|1.314	|1.292|
> |   | 5	|**0.920**   |1.447 |1.899 |1.436 |1.448 |	| 5	|**1.010**	|1.319	|1.338	|1.330	|1.306|
> |   | 6	|**0.939**   |1.496 |1.986 |1.506 |1.457 |	| 6	|**1.014**	|1.328	|1.356	|1.342	|1.314|
> |   | 7	|**0.958**   |1.529 |2.054 |1.558 |1.469 |	| 7	|**1.021**	|1.339	|1.374	|1.357	|1.326|
> |   | 8	|**0.977**   |1.552 |2.104 |1.598 |1.483 |	| 8	|**1.029**	|1.349	|1.390	|1.372	|1.335|
> |   | 9	|**0.984**   |1.570 |2.140 |1.612 |1.498 |	| 9	|**1.035**	|1.357	|1.402	|1.381	|1.343|
> |   | 10|**0.996**   |1.583 |2.177 |1.615 |1.507 |	| 10|**1.040**	|1.364	|1.412	|1.390	|1.351|
> |   | 11|**0.999**   |1.596 |2.226 |1.620 |1.507 |	| 11|**1.040**	|1.370	|1.421	|1.395	|1.358|
> |   | 12|**1.004**   |1.615 |2.284 |1.633 |1.505 |	| 12|**1.043**	|1.374	|1.432	|1.401	|1.364|
> |psl| 1 |**154.569** |192.223 |192.649 |189.818 |191.848 |z500	|1 |**19.833** |24.322 |24.825 |24.160 |24.566 |
> |   | 2	|**154.931** |201.041 |204.039 |199.346 |199.155 |	    |2 |**19.926** |25.503 |26.731 |25.988 |25.328 |
> |   | 3	|**157.615** |205.517 |210.173 |204.485 |201.779 |	    |3 |**20.377** |26.042 |27.453 |27.019 |25.655 |
> |   | 4	|**158.154** |208.316 |214.462 |207.659 |204.259 |	    |4 |**20.646** |26.359 |27.906 |27.630 |25.941 |
> |   | 5	|**159.407** |210.686 |217.911 |209.790 |205.218 |	    |5 |**21.007** |26.713 |28.175 |28.275 |26.113 |
> |   | 6	|**160.212** |211.672 |220.934 |212.426 |206.170 |	    |6 |**21.305** |26.960 |28.580 |29.056 |26.205 |
> |   | 7	|**161.927** |213.378 |223.726 |215.205 |208.116 |	    |7 |**21.748** |27.283 |29.026 |29.632 |26.484 |
> |   | 8	|**164.266** |215.256 |226.267 |217.126 |210.184 |	    |8 |**22.209** |27.514 |29.468 |30.007 |26.844 |
> |   | 9	|**164.532** |216.191 |227.955 |217.776 |211.274 |	    |9 |**22.499** |27.733 |29.770 |30.190 |27.163 |
> |   | 10|**166.283** |216.270 |230.086 |218.335 |210.944 |	    |10|**22.712** |27.818 |30.237 |30.287 |27.377 |
> |   | 11|**165.951** |216.072 |232.161 |219.141 |211.645 |	    |11|**22.756** |27.840 |30.860 |30.343 |27.565 |
> |   | 12|**165.857** |216.629 |232.499 |218.666 |211.306 |	    |12|**22.776** |27.929 |31.294 |30.365 |27.588 |
>
> We can find that ClimateAR outperforms all baselines with an average CRPS reduction of **23.33%** , demonstrating its effective ability to model the uncertainties of the climate system. These results are added in **Fig. 5 (Section 5.4)** and **Table 5 (Appendix A.4.1)** of our revised manuscript.
>
>
> (b) In **Fig.6 (Section 5.4)** of our revised manuscript, we conduct **probabilistic ENSO forecasting** experiments and show some ensemble members.
>
> - In the left figure, we compute the standard deviation of the ensemble forecast members and plot the ensemble mean together with the range defined by ± two standard deviations as the probabilistic forecast interval in 6-month lead time. We can find that the actual observations almost entirely fall within ClimateAR's confidence range, demonstrating that the model effectively captures the uncertainty of ENSO.
> - The right figure shows the distribution of each ensemble predictor member over the next 12 months when forecasts are initiated from different initial months, demonstrating that ClimateAR can stably track the development and decline of ENSO in a probabilistic sense. This further confirms that the model does not simply output a single deterministic value, but is able to characterize the probabilistic evolution path of ENSO, a typical climate mode.

---

> ### Author Response · Authors · 2025-11-22
> **Response to Reviewer 36PM (Part 2)**
>
> >Weakness 2 and Question 2：The problem needs to be reframed as "long-term weather forecasting" or the forecasting and evaluation framework needs to be adjusted according to the climate forecast problem. Also, the RMSE results show the performance tend to de grade after 8 months lead time on several variables.
>
> **Reformulate our framework and evaluation.**
>
> Thanks for the suggestion. We clarify that our framework is primarily designed for near‑term climate forecasting on the seasonal‑to‑interannual (S2I) timescale, which lies between short‑term weather forecasting (days to weeks) and long‑term climate change trends (over decades).
>
> - Our main evaluation window of 1–10 months lies squarely within the standard seasonal‑to‑interannual range. Many S2I forecasts works commonly use 1–12‑month lead times because this interval provides the strongest predictable signal and is most relevant for key climate modes such as ENSO.
> - Moreover, as shown in **Fig. 9 (Appendix A.4.1)** of the revised manuscript, we further extend ClimateAR’s forecast horizon to 1–14 months in regions with inherently higher predictability (low‑latitude regions), reaching the annual scale and remaining consistent with the S2I framework.
> - Regarding ACE2 and NeuralGCM (climate simulation part), they focus on reproducing changes in climate statistical characteristics on decades scale through externally forcing signals, rather than predicting specific (anomalous) variable values. This is different from our objective of predicting real anomalies on seasonal-to-interannual timescales.
>
> **Explore potential of ClimateAR for decadal climate simulation.**
>
> Thank you also for noting that RMSE degradation appears beyond 8‑month lead times for some variables. This is because the model is trained by single‑step, making it prone to climatology drift. For S2I forecasting, this degradation is acceptable and will not have a significant impact on forecasting skill on anomalies . However, when we further explore the potential of ClimateAR on decadal climate simulation, the degradation becomes unacceptable, which indicates the drift of climatology. To address this, we applied a simple rolling‑prediction fine‑tuning procedure on ERA5 with only a few additional epochs. This lightweight adjustment effectively mitigates long‑term RMSE drift. We plotted the  RMSE comparison for whether using rolling fine-tune in **Fig. 18 (Appendix A.5)** of our revised manuscript.
>
> Then, we conduct an experiment to further evaluate the climate simulation potential of ClimateAR over multi‑year to decadal timescales. As shown in **Fig. 7 (Section 5.5**) of the revised manuscript, we show the annual and monthly global‑mean 2‑metre temperature (t2m) from 1958–2020, using only the January 1958 initial state. From the result we can find that:
> - ClimateAR produces a stable and physically plausible long-term evolution of the temperature field over this multi-decadal period. This demonstrates that ClimateAR can maintain stability and capture long‑term climate trajectories over decadal periods.
> - It is worth noting that ACE2 explicitly incorporates various external forcing inputs such as CO2, SSTs and sea-ice, whereas ClimateAR does not include a dedicated interface for forcing signals and instead relies solely on SST updates as a forcing input. Remarkably, even under this simplified setting, ClimateAR aligns well with observed multi‑decadal climate statistics, highlighting its potential as a long‑term climate forecasting framework.
>
> Thank you for your construstive suggestion. Accordingly, we reformulate the problem solved in this paper as **seasonal-to-interannual climate forecasting**. In addition, to further explore ClimateAR’s potential for long‑term climate forecasting, we perform a simple fine‑tuning on the rolling‑prediction and conduct a simple long‑term climate simulation experiment following the setups of ACE2 and NeuralGCM. The results show that ClimateAR has potential to extend effectively to decadal forecasting.

---

> ### Author Response · Authors · 2025-11-28
> **Looking forward to your feedback**
>
> Dear Reviewer,
>
> Thank you again for your valuable review, which has greatly inspired us to improve our paper. This is a kind reminder that only a few days remain in the discussion period. We kindly ask if our responses have addressed your concerns. If you have any further questions, we would be more than willing to engage in further discussions and make any necessary improvements.
>
> Thank you once again for dedicating your valuable time to reviewing our work. We look forward to your feedback.

---

### Official Review · Reviewer_r9QE · 2025-10-27

**Soundness:** 3
**Presentation:** 3
**Contribution:** 3
**Rating:** 4
**Confidence:** 4

**Summary:**

This paper adapts a visual autoregressive (AR) model for probabilistic climate forecasting with two key innovations. First, an aligned tokenizer with a "shallow-separation, deep-sharing" architecture bridges the domain gap between simulated and real-world data. Second, a mixed-scale conditioning mechanism effectively handles high-dimensional climate inputs by combining global and local guidance.

**Strengths:**

This paper introduce an innovative approach to tackle to problem of long term predictions. In addition investigate the domain shift between simulated and real-world data. The model achieves impressive results, showing a significant average improvement in ACC over tested baselines.

**Weaknesses:**

- A central claim of the paper is the model's superiority in probabilistic forecasting and capturing climate uncertainty. However, the primary evaluation relies on the Anomaly Correlation Coefficient (ACC), a deterministic metric that assesses the capability of the ensemble mean. This metric does not evaluate the quality of the predicted probability distribution itself (e.g., its spread or calibration). To fully substantiate the claims of probabilistic skill, the inclusion of a proper probabilistic metric, such as the Continuous Ranked Probability Score (CRPS) and Spread/Skill-Ratio would be essential.
-  While the paper provides an efficiency study for model training, it omits a detailed analysis of the inference cost. Key details regarding the inference time and computational resources required to generate a full ensemble forecast are missing. This information is critical for assessing the model's practical viability for operational applications, where computational efficiency is often a major constraint. In addition it would be helpful to have a better understanding if the gain obtained from the new architecture comes from a higher complexity compared to the baseline.

**Questions:**

- The VQ tokenizer is a critical component of ClimateAR, yet its design choices lack a detailed ablation study. Specifically:
a) Codebook Utilization: The paper does not provide an analysis of the codebook utilization or perplexity. This is important, as low utilization (i.e., "codebook collapse") can indicate an inefficient latent space where only a fraction of the learned codes are used, suggesting the codebook may be unnecessarily large.
b) Codebook Size: While the authors provide a hyperparameter study for the number of codebooks (partitions N), there is no corresponding study on the size of the codebooks (V, set to 4096). An ablation on this key hyperparameter would be needed to understand the trade-offs between representational capacity and model efficiency.
- The paper claims probabilistic superiority but primarily uses a deterministic metric (ACC of the ensemble mean). Could the authors provide results using a proper probabilistic score, such as the Continuous Ranked Probability Score (CRPS), to validate this core claim?
- How does the model ensure that the generated token sequences decode into physically plausible climate states, especially concerning conservation laws? Was any analysis performed to verify this?
- What is the practical inference cost to generate a full 200-member ensemble forecast, and how does this compare to the operational baselines?
- Could the authors provide a targeted ablation study to isolate the impact of the hybrid-scale prompt (C_mix) on the model's overall performance?
- Given the widespread success of diffusion models in scientific generative modeling, the paper lacks a compelling justification for choosing an autoregressive framework. Could the authors either include a state-of-the-art conditional diffusion model as a baseline or provide a more rigorous argument for why the AR paradigm is fundamentally better suited for long-range climate forecasting?

---

> ### Author Response · Authors · 2025-11-22
> **Response to Reviewer r9QE (Part 1)**
>
> We sincerely thank you for your constructive comments of our work. We have carefully considered your concerns regarding ablation studies, computational cost, and probabilistic forecasting. Below, we provide detailed responses and new experimental evidence to address these points.
>
> >Question 1: The VQ tokenizer lacks a detailed ablation study.
>
> Thank you for pointing out the lack of a detailed ablation study for the VQ tokenizer. We fully agree that codebook size is a key hyperparameter of ClimateAR to understand the trade-offs between representational capacity and model efficiency. Therefore, we conducted a hyperparameter study on the codebook size (V) **in Table 8 (Appendix A.4.6)**. The results of ACCs with different codebook sizes (V) during the pre-training phase are as follows:
>
> |ACC|1024|2048	|4096	|8192|
> | :---: | :---: | :---: | :---: | :---: |
> |z500|	0.402|	0.408|	**0.428**|	0.397|
> |tas|	0.463|	0.467|	**0.480**|	0.466|
> |r*($\times 10^-5$)	|0.164	|0.162|	**0.190**|	0.161|
> |psl|	0.350	|0.350	|**0.374**	|0.330|
>
> Beside, we fully agree that codebook utilization is a crucial diagnostic, and low utilization (i.e., codebook collapse) is unacceptable. In fact, our experiments have confirmed that the **utilization rate** consistently approaches **100% in all settings**. This demonstrates that our VQ tokenizer can learn a rich and efficient latent space, thereby effectively distinguishing climate patterns that are inherently similar in structure and shape.
>
> >Question 2 and Weakness 1: Could the authors provide results using a probabilistic score, such as CRPS.
>
> Thank you for suggesting the use of proper probabilistic scores, such as the **continuous ranked probability score (CRPS)**. We evaluated the CRPS results for ClimateAR and all the baselines. For the deterministic baseline, we obtained different ensemble forecast members through slight perturbations in the initial field. The results are as follows:
>
> |CRPS|lead	|ClimateAR	|GraphCast|	Pangu	|Oneforecast	|ClimaX	|CRPS |lead	|ClimateAR	|GraphCast|	Pangu	|Oneforecast	|ClimaX	|
> | :---:  |  :---:  | :---:  |  :---:  | :---:  |  :---:  | :---:  | :---:  |  :---:  | :---:  |  :---:  | :---: | :---: | :---: |
> |t2m|	1	|**0.831** |0.929 |1.198 |0.901 |1.293 |pr $(\times 10^{-5})$| 1|**1.005**|1.205|	1.209|1.211|1.222|
> |   | 2	|**0.847**   |1.152 |1.463 |1.118 |1.371 |	| 2	|**1.000**	|1.256	|1.258	|1.265	|1.256|
> |   | 3	|**0.878**   |1.293 |1.650 |1.263 |1.418 |	| 3	|**1.002**	|1.285	|1.290	|1.295	|1.275|
> |   | 4	|**0.900**   |1.383 |1.790 |1.360 |1.439 |	| 4	|**1.008**	|1.304	|1.314	|1.314	|1.292|
> |   | 5	|**0.920**   |1.447 |1.899 |1.436 |1.448 |	| 5	|**1.010**	|1.319	|1.338	|1.330	|1.306|
> |   | 6	|**0.939**   |1.496 |1.986 |1.506 |1.457 |	| 6	|**1.014**	|1.328	|1.356	|1.342	|1.314|
> |   | 7	|**0.958**   |1.529 |2.054 |1.558 |1.469 |	| 7	|**1.021**	|1.339	|1.374	|1.357	|1.326|
> |   | 8	|**0.977**   |1.552 |2.104 |1.598 |1.483 |	| 8	|**1.029**	|1.349	|1.390	|1.372	|1.335|
> |   | 9	|**0.984**   |1.570 |2.140 |1.612 |1.498 |	| 9	|**1.035**	|1.357	|1.402	|1.381	|1.343|
> |   | 10|**0.996**   |1.583 |2.177 |1.615 |1.507 |	| 10|**1.040**	|1.364	|1.412	|1.390	|1.351|
> |   | 11|**0.999**   |1.596 |2.226 |1.620 |1.507 |	| 11|**1.040**	|1.370	|1.421	|1.395	|1.358|
> |   | 12|**1.004**   |1.615 |2.284 |1.633 |1.505 |	| 12|**1.043**	|1.374	|1.432	|1.401	|1.364|
> |psl| 1 |**154.569** |192.223 |192.649 |189.818 |191.848 |z500	|1 |**19.833** |24.322 |24.825 |24.160 |24.566 |
> |   | 2	|**154.931** |201.041 |204.039 |199.346 |199.155 |	    |2 |**19.926** |25.503 |26.731 |25.988 |25.328 |
> |   | 3	|**157.615** |205.517 |210.173 |204.485 |201.779 |	    |3 |**20.377** |26.042 |27.453 |27.019 |25.655 |
> |   | 4	|**158.154** |208.316 |214.462 |207.659 |204.259 |	    |4 |**20.646** |26.359 |27.906 |27.630 |25.941 |
> |   | 5	|**159.407** |210.686 |217.911 |209.790 |205.218 |	    |5 |**21.007** |26.713 |28.175 |28.275 |26.113 |
> |   | 6	|**160.212** |211.672 |220.934 |212.426 |206.170 |	    |6 |**21.305** |26.960 |28.580 |29.056 |26.205 |
> |   | 7	|**161.927** |213.378 |223.726 |215.205 |208.116 |	    |7 |**21.748** |27.283 |29.026 |29.632 |26.484 |
> |   | 8	|**164.266** |215.256 |226.267 |217.126 |210.184 |	    |8 |**22.209** |27.514 |29.468 |30.007 |26.844 |
> |   | 9	|**164.532** |216.191 |227.955 |217.776 |211.274 |	    |9 |**22.499** |27.733 |29.770 |30.190 |27.163 |
> |   | 10|**166.283** |216.270 |230.086 |218.335 |210.944 |	    |10|**22.712** |27.818 |30.237 |30.287 |27.377 |
> |   | 11|**165.951** |216.072 |232.161 |219.141 |211.645 |	    |11|**22.756** |27.840 |30.860 |30.343 |27.565 |
> |   | 12|**165.857** |216.629 |232.499 |218.666 |211.306 |	    |12|**22.776** |27.929 |31.294 |30.365 |27.588 |
>
>
> We can observe that ClimateAR outperforms all baselines with an average CRPS reduction of **23.33%** , demonstrating its effective ability to model the uncertainties of the climate system. These results are added in **Fig. 5 (Section 5.4) and Table 5 (Appendix A.4.1)** of our revised manuscript.

---

> ### Author Response · Authors · 2025-11-22
> **Response to Reviewer r9QE (Part 2)**
>
> >Question 3: How does the model ensure that the generated token sequences decode into physically plausible climate states.
>
> Thank you for raising this important question regarding physical plausibility in the decoded climate states. While ClimateAR does not explicitly enforce physical equations, it benefits from a **fully data‑driven approach** grounded in physically consistent training data. In particular, the model is first **pre‑trained on large‑scale simulation data** generated by physics‑based models, which inherently **satisfy conservation laws** and fundamental dynamical constraints. Through this pre-training, ClimateAR implicitly learns these physical relationships and structural regularities and transfers these learned physical laws to real-world data. As demonstrated in our visualization and ENSO prediction experiments, the resulting decoded fields maintain realistic spatial patterns, temporal evolution, and large‑scale climate modes, indicating that the learned token representations preserve physically plausible climate behavior.
>
> >Weakness 2 and question 4: What is the practical inference cost to generate ensemble forecast and how does this compare to the operational baselines?
>
> Thank you for your insightful question regarding the practical inference cost of ensemble forecasting. ClimateAR's inference cost is $O(h^2 w^2)$, where $(h, w)$ is the size of feature map $\mathbf{f}$, consistent with its training complexity, and we further accelerate inference using a key-value cache. Besides theoretical efficiency, We supplement this with experiments on inference cost, the results of which are as follows:
>
> |Method |Parameters| GPU Memory(GB) |FLOPs |Training Time Min/Epoch|Inference Time s/Step| ACC|
> |:---:|:---:|:---:|:---:|:---:|:---:|:---:|
> |ClimaX| 104.0M| 19.27| 420.97G| 18.78| 0.162| 0.376|
> |Oneforecast| 24.7M| 38.95 |548.78G| 69.73 |0.516 |0.386|
> |GraphCast |28.9M| 34.59| 1852.26G| 32.33 |0.279| 0.376|
> |Pangu |23.9M| 7.60 |213.82G| 5.48| 0.047| 0.329|
> |ClimateAR |480.6M |18.34| 324.23G |13.13 |0.179| 0.464|
>
> As shown in the table, ClimateAR’s single‑member inference time and FLOPs are **lower than those of graph‑based baseline and comparable to the ViT‑based baselines**. For ensemble forecasting, the total cost scales **linearly** with the number of ensemble members $M$ as each member requires an independent probabilistic sampling pass. Deterministic models would similarly incur an additional **$M$ times inference cost** when producing $M$ ensemble members. In our setting, with 200 ensemble members, ClimateAR achieves an inference time of only $0.179$s$\times200=35.8$s per month, which remains both practical and efficient for seasonal-to-decadal ensemble prediction.
>
> We add the inference cost results in **Table 8 (Appendix A.4.7)** of our revised manuscript.
>
> >Question 5: Could the authors provide a targeted ablation study to isolate the impact of the hybrid-scale prompt ($\mathbf{C} _ {mix}$) on the model's overall performance?
>
> Thank you for recommending a targeted ablation study to isolate the impact of the hybrid-scale prompt ($\mathbf{C} _ {mix}$). For the ablation experiment of $\mathbf{C} _ {mix}$, we have already given it in **Section 5.6**. We removed the mix prompt separately without modifying any other modules or parameters, ensuring isolated operation of this module. The results show that the model using $\mathbf{C} _ {mix}$ achieves higher averaged ACCs for 6 months prediction.
> |ACC|	ClimateAR	|w/o prompt|
> |:---:|:---:|:---:|
> |z500|	**0.147**	|0.141 |
> |t2m|	**0.212** 	|0.186 |
> |pr	|**0.173** 	|0.164 |
> |psl|	**0.228**| 	0.224 |
>
>
> >Question 6: Could the authors either include a state-of-the-art conditional diffusion model as a baseline or provide a more rigorous argument for why the AR paradigm is fundamentally better suited for long-range climate forecasting?
>
> Thank you for your valuable suggestion to compare against a state-of-the-art conditional diffusion model or better justify the AR paradigm. We would provide a brief justification to explain why AR modeling is particularly well suited for long-range climate forecasting: Existing state-of-the-art diffusion-based models, such as GenCast, are designed for short-term weather forecasting and cannot be extended for long-range climate forecasting. This is because long-range climate predictability arises from slowly evolving large-scale phenomena such as ENSO, while existing diffusion-based climate models generally treat the climate system as a monolithic stochastic field and **fail to explicitly model these structured multi-scale dependencies**. However, ClimateAR effectively addresses physical phenomena such as multi-scale energy cascades in the climate system through multi-scale modeling, which is more suitable for long-range climate forecasting.

---

> ### Author Response · Authors · 2025-11-28
> **Looking forward to your feedback**
>
> Dear Reviewer,
>
> Thank you again for your valuable review, which has greatly inspired us to improve our paper. This is a kind reminder that only a few days remain in the discussion period. We kindly ask if our responses have addressed your concerns. If you have any further questions, we would be more than willing to engage in further discussions and make any necessary improvements.
>
> Thank you once again for dedicating your valuable time to reviewing our work. We look forward to your feedback.

---

### Official Review · Reviewer_C4po · 2025-10-29

**Soundness:** 3
**Presentation:** 4
**Contribution:** 3
**Rating:** 8
**Confidence:** 4

**Summary:**

The paper proposes a visual autoregressive model for climate forecasting. Given one timestep of climate model data, it predicts a distribution over the next timestep. It computes tokens corresponding to multiple resolutions of the data, and uses a transformer to predict the tokens at the next timestep,  before decoding the tokens in observations space, effectively capturing dynamics happening at multiple spatial scales.

Authors evaluate the model on ERA5 and ORA5 data, and show that the model outperform all baselines.

**Strengths:**

The paper is sound and well written. Using a visual autoregressive model to capture multi-scale dependencies is a great idea, and the model also allows to represent uncertainty, although this is not explored nor validated by experiments.

**Weaknesses:**

The experiments and results could be strengthened.

First, the RMSE is not a great metric for climate emulators. When doing climate forecasting, the variables are not predictable and we're instead interested in capturing climate statistics and dynamics. It is thus important to report additional metrics, such as mean + std. dev., or the power spectral density of the climate indices (as done in [1]) or the time-dependent area-weighted global mean and the area-weighted global mean bias and RMSE of time-mean fields (as done in [2]).

Second, it would be interesting to highlight learned relationships between multi-scale processes. For example by computing regression coefficient between climate indices (such as ENSO or IOD) with variables over the entire grid.

Third, authors claim that the model does probabilistic climate forecasting, but this is not shown or evaluated in any of the experiments. Does the model learn a sensible uncertainty representation? I believe that the authors need to evaluate it since they argue that it is a strength of the model. To do so, it might be useful to evaluate the model on climate model data, where there are many more available years  than ERA5. Authors could show the uncertainty in the climate forecast (does it increase as you forecast longer time horizons?), continuous ranked probability score (by looking at different initial conditions for the different models) and maps of the uncertainty to check if predictable processes are associated with less uncertainty (are oceanic temperatures associated with less uncertainty than atmospheric temperatures over land?).

The authors should also highlight the limitations of the model in the discussion section.

[1] Hickman et al., Causal Climate Emulation with Bayesian Filtering, 2025

[2] Watt-Meyer et al., ACE: A fast, skillful learned global atmospheric model for climate prediction, 2023

**Questions:**

The Intra-scale Mixed Token paragraph is a bit unclear to me.

In Equation 8, why are you doing down(f')? Isn't f'k already at the correct resolution?

Why does it help to do the intra scale mized token? Since the "f" is essentially a concatenation of the r (from Fig. 1 a), and this is replacing the r with the f.

You're then adding all r' into Cmix, but r' for low k are already contained in the Intra-scale Mixed Token. Isn't this information contained twice then? Or is Cmix only looking at higher resolution tokens?

---

> ### Author Response · Authors · 2025-11-22
> **Response to Reviewer C4po (Part 1)**
>
> We sincerely thank you for your constructive comments of our work. We have carefully considered your concerns regarding
>  probabilistic forecasting and evaluation metrics. Below, we provide detailed responses and new experimental evidence to address these points.
>
> >Weakness 1: It would be interesting to highlight learned relationships between multi-scale processes. For example by computing regression coefficient between climate indices (such as ENSO or IOD) with variables over the entire grid.
>
> Thank you for this valuable suggestion. In our manuscript, we already highlight the learned relationships related to multi‑scale climate processes by computing climate indices in **Section 5.3**. Specifically, we compute the **ENSO index** using the mean SST over the **entire Niño 3.4 grid** and evaluate the model’s ENSO forecasting skill through the **Correlation Coefficient**. The detailed forecasting skills of the ENSO index are as follows:
>
> |lead| ClimateAR|	ClimaX|	Pangu|	GraphCast|	Oneforecast	|Dwd|
> | :---:  |  :---:  | :---:  |  :---:  | :---:  |  :---:  | :---:  |
> 1	|**0.968**|	0.95	|0.954	|0.953	|0.971	|0.904|
> 2	|**0.939**|	0.918	|0.908	|0.882	|0.920	|0.834|
> 3	|**0.892**|	0.864	|0.844	|0.774	|0.852	|0.749|
> 4	|**0.848**|	0.797	|0.747	|0.647	|0.766	|0.700|
> 5	|**0.803**|	0.737	|0.640	|0.496	|0.650	|0.611|
> 6	|**0.762**|	0.649	|0.533	|0.346	|0.540
> 7	|**0.733**|	0.516	|0.400	|0.220	|0.400
> 8	|**0.725**|	0.369 |	0.296	|0.097	|0.315
> 9	|**0.664**|	0.234 |	0.184	|-0.006	|0.194
> 10	|**0.606**|	0.130 |	0.096	|-0.088	|0.084
> 11	|**0.593**|	0.077 |	0.038	|-0.109	|0.040
> 12	|**0.620**|	0.067 |	0.021	|-0.100	|0.051
>
> Specifically, ClimateAR can maintain a high ENSO skill (>0.6) over a 12-month forecast horizon, outperforming all the baselines. These results demonstrate that our model captures key large‑scale climate variability.

---

> ### Author Response · Authors · 2025-11-22
> **Response to Reviewer C4po (Part 2)**
>
> >Weakness 2: This is not shown or evaluated in any of the experiments for probabilistic climate forecasting.
>
> Thank you for suggesting adding probabilistic forecasting experiments. We have provided two important experiments to demonstrate this:
>
> (a) We evaluated the **continuous ranked probability score (CRPS)**. For the deterministic baseline, we obtained different ensemble forecast members through slight perturbations in the initial field. The detail results of CRPS are as follows:
>
> |CRPS|lead	|ClimateAR	|GraphCast|	Pangu	|Oneforecast	|ClimaX	|CRPS |lead	|ClimateAR	|GraphCast|	Pangu	|Oneforecast	|ClimaX	|
> | :---:  |  :---:  | :---:  |  :---:  | :---:  |  :---:  | :---:  | :---:  |  :---:  | :---:  |  :---:  | :---: | :---: | :---: |
> |t2m|	1	|**0.831** |0.929 |1.198 |0.901 |1.293 |pr $(\times 10^{-5})$| 1|**1.005**|1.205|	1.209|1.211|1.222|
> |   | 2	|**0.847**   |1.152 |1.463 |1.118 |1.371 |	| 2	|**1.000**	|1.256	|1.258	|1.265	|1.256|
> |   | 3	|**0.878**   |1.293 |1.650 |1.263 |1.418 |	| 3	|**1.002**	|1.285	|1.290	|1.295	|1.275|
> |   | 4	|**0.900**   |1.383 |1.790 |1.360 |1.439 |	| 4	|**1.008**	|1.304	|1.314	|1.314	|1.292|
> |   | 5	|**0.920**   |1.447 |1.899 |1.436 |1.448 |	| 5	|**1.010**	|1.319	|1.338	|1.330	|1.306|
> |   | 6	|**0.939**   |1.496 |1.986 |1.506 |1.457 |	| 6	|**1.014**	|1.328	|1.356	|1.342	|1.314|
> |   | 7	|**0.958**   |1.529 |2.054 |1.558 |1.469 |	| 7	|**1.021**	|1.339	|1.374	|1.357	|1.326|
> |   | 8	|**0.977**   |1.552 |2.104 |1.598 |1.483 |	| 8	|**1.029**	|1.349	|1.390	|1.372	|1.335|
> |   | 9	|**0.984**   |1.570 |2.140 |1.612 |1.498 |	| 9	|**1.035**	|1.357	|1.402	|1.381	|1.343|
> |   | 10|**0.996**   |1.583 |2.177 |1.615 |1.507 |	| 10|**1.040**	|1.364	|1.412	|1.390	|1.351|
> |   | 11|**0.999**   |1.596 |2.226 |1.620 |1.507 |	| 11|**1.040**	|1.370	|1.421	|1.395	|1.358|
> |   | 12|**1.004**   |1.615 |2.284 |1.633 |1.505 |	| 12|**1.043**	|1.374	|1.432	|1.401	|1.364|
> |psl| 1 |**154.569** |192.223 |192.649 |189.818 |191.848 |z500	|1 |**19.833** |24.322 |24.825 |24.160 |24.566 |
> |   | 2	|**154.931** |201.041 |204.039 |199.346 |199.155 |	    |2 |**19.926** |25.503 |26.731 |25.988 |25.328 |
> |   | 3	|**157.615** |205.517 |210.173 |204.485 |201.779 |	    |3 |**20.377** |26.042 |27.453 |27.019 |25.655 |
> |   | 4	|**158.154** |208.316 |214.462 |207.659 |204.259 |	    |4 |**20.646** |26.359 |27.906 |27.630 |25.941 |
> |   | 5	|**159.407** |210.686 |217.911 |209.790 |205.218 |	    |5 |**21.007** |26.713 |28.175 |28.275 |26.113 |
> |   | 6	|**160.212** |211.672 |220.934 |212.426 |206.170 |	    |6 |**21.305** |26.960 |28.580 |29.056 |26.205 |
> |   | 7	|**161.927** |213.378 |223.726 |215.205 |208.116 |	    |7 |**21.748** |27.283 |29.026 |29.632 |26.484 |
> |   | 8	|**164.266** |215.256 |226.267 |217.126 |210.184 |	    |8 |**22.209** |27.514 |29.468 |30.007 |26.844 |
> |   | 9	|**164.532** |216.191 |227.955 |217.776 |211.274 |	    |9 |**22.499** |27.733 |29.770 |30.190 |27.163 |
> |   | 10|**166.283** |216.270 |230.086 |218.335 |210.944 |	    |10|**22.712** |27.818 |30.237 |30.287 |27.377 |
> |   | 11|**165.951** |216.072 |232.161 |219.141 |211.645 |	    |11|**22.756** |27.840 |30.860 |30.343 |27.565 |
> |   | 12|**165.857** |216.629 |232.499 |218.666 |211.306 |	    |12|**22.776** |27.929 |31.294 |30.365 |27.588 |
>
> We can find that ClimateAR outperforms all baselines with an average CRPS reduction of **23.33%** , demonstrating its effective ability to model the uncertainties of the climate system. These results are added in **Fig. 5 (Section 5.4)** and **Table 5 (Appendix A.4.1)** of our revised manuscript.
>
>
> (b) In **Fig.6 (Section 5.4)** of our revised manuscript, we conduct **probabilistic ENSO forecasting** experiments.
>
> - In the left figure, we compute the standard deviation of the ensemble forecast members and plot the ensemble mean together with the range defined by ± two standard deviations as the probabilistic forecast interval in 6-month lead time. We can find that the actual observations almost entirely fall within ClimateAR's confidence range, demonstrating that the model effectively captures the uncertainty of ENSO.
>
> - The right figure shows the distribution of each ensemble predictor member over the next 12 months when forecasts are initiated from different initial months, demonstrating that ClimateAR can stably track the development and decline of ENSO in a probabilistic sense. This further confirms that the model does not simply output a single deterministic value, but is able to characterize the probabilistic evolution path of ENSO, a typical climate mode. As the ensemble mean is used as the input for the next step in autoregressive prediction, we can observe that the variance does not increase substantially as the forecast horizon grows, and the long‑term forecast is stable.

---

> ### Author Response · Authors · 2025-11-22
> **Response to Reviewer C4po (Part 3)**
>
> >Weakness 3: RMSE is not a great metric for climate emulators. When doing climate forecasting, the variables are not predictable and we're instead interested in capturing climate statistics and dynamics.  It is thus important to report additional metrics, such as mean + std, or the power spectral density of the climate indices or the time-dependent area-weighted global mean and the area-weighted global mean bias and RMSE of time-mean fields.
>
> Thank the reviewer for the insightful comment. We fully agree that RMSE is insufficient for evaluating climate emulators or probabilistic climate forecasting models. Climate forecasting emphasizes the ability to represent climate statistics, variability, uncertainty, and dynamical modes, rather than matching individual variable values. In the revised manuscript, we highlight that our work already evaluates these richer climate-statistical metrics, and we further clarify and expand these results. Specifically:
>
> - We use **ACC** as the principal metric for assessing the forecasting skill for seasonal-to-decadal climate forecasting. This can quantify the spatial and temporal agreement between predicted and observed anomalous deviations from the long-term climatological mean.
>
> - For climate statistics and dynamics (such as **mean+std**), **in Fig. 6 (Section 5.4)** of our revised manuscript, the probabilistic ENSO forecasting explicitly reports the ensemble mean and ensemble spread (±2 std), demonstrating that ClimateAR captures both the central tendency and variability of key climate indices.
>
> - For **power spectral density** of the climate indices (as done in [1]), we conduct an experiment on the Niño3.4 index **in Fig. 14 (Appendix A.4.3)** of our revised manuscript. Specifically, we compute the PSD of the Niño3.4 index time series generated by ClimateAR at 6-month lead time and compare it with the observation. The PSD results show that ClimateAR successfully captures the primary ENSO peak at interannual timescales. However, we also observe two systematic deviations. First, the high-frequency components exhibit slightly elevated power compared with the reference, suggesting that the model produces somewhat stronger variability. Second, the overall energy level is higher, reflecting that the predicted Niño3.4 index tend to have larger amplitude than observation. This likely arises from the probabilistic modeling of ClimateAR, which inherently preserves more variability and thus increases high‑frequency power and amplitude.
>
> - For the **time-dependent area-weighted global mean** (as done in [2]), **in Fig. 7 (Section 5.5)** of our revised manuscript, we report the long-term evolution of annual and monthly area-weighted global mean 2 metre temperature (t2m) from 1958–2020 with initial state input of January 1958 and SST forcing input. This evaluates the model’s ability to preserve climate evolution across decades.
>
> [1] Hickman et al., Causal Climate Emulation with Bayesian Filtering, 2025.
>
> [2] Watt-Meyer et al., ACE: A fast, skillful learned global atmospheric model for climate prediction, 2023.

---

> ### Author Response · Authors · 2025-11-22
> **Response to Reviewer C4po (Part 4)**
>
> >Question 1: In Equation 8, why are you doing down(f')? Isn't f'k already at the correct resolution?
>
> Thank you for raising this point. In Eq.2, $\tilde{\mathbf{f}} _ k = \sum _ {i=1}^k \mathrm{up}({\mathbf r} _ i,(h,w))$, when we obtain $\tilde{\mathbf{f}} _ k$, we interpolate all ${\mathbf r} _ {<k}$ to the maximum size to facilitate the summation of these residuals ${\mathbf r} _ {<k}$. Therefore, although the size of ${\mathbf r} _ {<k}$ is different for each scale, the corresponding feature map $\tilde{\mathbf{f}} _ k$ has the same maximum size ($h,w$). Then, when constructing the Intra-scale Mixed Token, we need to downscale the feature $\tilde{\mathbf{f}} _ k$ ($h,w$) to the same size as the predicted content ${\hat {\mathbf r}} _ k$ ($h _ k, w _ k$).
>
> >Question 2: Why does it help to do the intra-scale mixed token？
>
> This is indeed a good question. In the early explorations of this work, we tried only using all the conditional tokens as the prompt prefix (which is also presented in the ablation study), but we found that overly complex conditional information about the climate system made it difficult for the model to converge. However, constructing intra-scale mixed tokens can effectively capture the dependencies between two consecutive time steps at the same scale. This approach can break down complex conditional inputs into multi-scale corresponding conditional inputs, effectively reducing the difficulty of extracting and modeling conditional information.
>
> >Question 3: You're then adding all $r^\prime$ into $\mathbf{C} _ {mix}$, but $r^\prime$ for low $k$ are already contained in the Intra-scale Mixed Token. Isn't this information contained twice then? Or is $\mathbf{C} _ {mix}$ only looking at higher resolution tokens?
>
> In autoregression, processes at scale $k$ cannot utilize the conditional information from higher resolution Intra-scale Mixed Token ${\mathbf r}^\prime _ {>k}$, This will result in the loss of some conditional information. Therefore, we introduced $\mathbf{C} _ {mix}$ to provide a global correction. However, since $\mathbf{C} _ {mix}$ provides global information for the generation of all the scales, it inevitably contains twice low scale information. Although low-resolution information ${\mathbf r}^\prime _ {<k}$ is duplicated among the 256 tokens in $\mathbf{C} _ {mix}$, the model calculates attention scores between each token in $\mathbf{C} _ {mix}$ and ${\mathbf R} _ k$, so that each scale can adaptively utilize more relevant information from $\mathbf{C} _ {mix}$.

---

> > ### Comment · Reviewer_C4po · 2025-11-24
> >
> > I thank the reviewers for their reponses.
> >
> > The addition of CRPS + additional evaluation methods strengthen the evaluation of the model.
> >
> > About the "learned relationships between multi-scale processes", this is not shown by the ENSO index forecast experiment.
> > What I asked was about studying the links between global and local processes, for examplke between GMST and ENSO. In other words, is ClimateAR able to capture relationships between multi-sclae processes?
> >
> > Also, for evaluating the learned probability distribution, it would be good to add standard deviation in Figure 14, and also add additional climate indices?

---

> ### Author Response · Authors · 2025-11-26
> **Response to Reviewer C4po**
>
> We sincerely thank you for your constructive comments on our work and insightful suggestions. We have carefully considered your concerns regarding muti-scale processes relationship and adding probability distribution in Fig.14. Below, we provide detailed responses and more experimental evidence to address these points.
>
> >The ability of ClimateAR to capture multi‑scale relationships.
>
> Thank you for your patience in reading and pointing out any shortcomings in our response. We realize that a simple ENSO forecasting experiment alone does not demonstrate interactions across different scales. To demonstrate the interaction between different scales, we conduct two sets of analyses.
> - **Global and regional.** We plot the prediction (ahead 1 month) and observation for the **global mean surface temperature (GMST)** and **ENSO** time series in **Fig.15 (Appendix 4.4)** in our revised manuscript. We can find that the predicted time series successfully reproduces both the amplitude and the **lagged phase relationship** (3–6 months consistent with the atmosphere-ocean heat exchange processes.) observed in the real climate system. Besides, we compute the regression coefficient between ENSO and the global mean surface temperature (GMST) anomaly. The **regression coefficient** between ENSO and GMST predicted by ClimateAR is **0.131**, which closely matches those derived from ERA5 (**difference <  $0.05 ^\circ C$**). These results demonstrate that ClimateAR effectively captures global‑scale responses to ENSO.
>
> - **Regional and local.** To further demonstrate the multiscale capability, we compare the spatial fields of regression coefficients between ENSO and precipitation/t2m anomalies in **Fig.16 (Appendix 4.4)**  in our revised manuscript.  We can observe a high degree of spatial consistency between the observed and predicted distribution maps, and ClimateAR accurately captures the positive or negative pixel-wise responses to ENSO intensity. This demonstrates that the model is able to represent pixel-wise local variability associated with ENSO.
>
> Overall, GMST represents **large-scale global variability**, ENSO index represents **mid-scale variability**, and pixel‑wise precipitation/t2m anomalies represent **small‑scale variability**. The high consistency between predicted and observed regression coefficients across all these scales indicates that ClimateAR effectively models the interactions among climate processes and successfully captures cross‑scale variability.
>
> > Evaluating the learned probability distribution in Fig. 14.
>
> Thank you for your valuable suggestions. To evaluate the learned probability distribution, We added a corresponding interval of **twice the standard deviation** for the ENSO power spectral density (PSD) analysis series in **Fig.14** in the latest revised manuscript. In addition, to further demonstrate the validity of the experiment, we also added the same PSD analysis experiment for the **Indian Ocean Dipole (IOD)** index at 1 month ahead forecasting, and similarly added a range of two standard deviations. The results shows the ability of ClimateAR to model the probabilistic distribution of climate indices.

---

### Author Response · Authors · 2025-11-24
**Summary of Revisions and Responses**

We thank the reviewers for their valuable feedback. The reviewers acknowledge the paper is **sound, well-written**, and innovative to **capture multi-scale climate dependence** by employing the autoregressive model. They highlight the strengths in addressing **domain transferring**, **modeling uncertainty**, and significantly **improving ACCs (29.27%)** compared to baselineses. **Aligned tokenizer and mixed-scale conditioning** are creative and effective design methods. They also note the **robustness** of ClimateAR, demonstrated through zero-shot forecasting, ENSO forecasting, and ablation study. Beyond these strengths, the core concerns raised by the reviewers are summarized as follows:

- **Reviewer C4po, r9QE, 36PM** all suggest a probabilistic climate forecasting experiment to verify that the model learns the uncertainty of the climate system.

- **Reviewer C4po** is interested in further evaluating the model’s ability to capture multi‑scale climate relationships.

- **Reviewer C4po** is interested in how the model captures climate statistics and dynamics.

- **Reviewer r9QE** requests for additional ablation studies and efficiency evaluations.

- **Reviewer r9QE** requests for a clearer explanation of how the model captures physical relationships and why the AR generative paradigm is better than diffusion models.

- **Reviewer 36PM** requests reformulating the framework and evaluation to align more clearly with climate forecasting.


In response, we have substantially strengthened our manuscript through extensive additional experiments, expanded model analysis, and the framework reformulation. Key revisions of our manuscript and responses include:

- **More Empirical Studies on Probabilistic Forecasting**:

  - We add full **continuous ranked probability score (CRPS)** to evaluate the ability for uncertainty modeling, showing a **23.33%** average improvement compared with all baselines.

  - We add probabilistic ENSO forecasting with **ensemble spread and distributions** (mean ± 2std).

- **Analysis of the Relationship of Muti-scale Processes**:

  - We analyze the relationship of global mean surface temperature (GMST) and ENSO by comparing predicted and observed time series and computing their regression coefficients (**difference < $0.05^{\circ} C$**) to show the global ENSO responses.

  - We plot the spatial distribution of the regression coefficients between the ENSO index and pixel-wise pr/t2m to show local ENSO responses.

- **Climate‑statistics and Long‑term Simulation Analysis**:

  - We add **probabilistic power spectral density (PSD)** analysis of Niño3.4 and IOD indices to evaluate the modeling climate periods.

  - We add annual and monthly global‑mean t2m evolution over **1958–2020** to simulate the climatology change in decadal timescale.

- **More Ablations and Efficiency study**:

  - We add a hyperparameter study for the codebook size V (**optimum size 4096**) and analyze the codebook utilization (**nearly 100%**).

  - We re-explain the ablation of the hybrid-scale prompt by showing the ACC improvements (**11.60%**) of this key component.

  - We add a comparison of inference cost (**0.179s per step**) among different models to demonstrate competitive efficiency and practical ensemble runtime of ClimateAR.

- **Additional Theoretical Analysis**：
  - We explain that ClimateAR learns **data-driven physical consistency** from physics‑based simulation data because its tokenizer design and training paradigm provide strong transferability, enabling the model to effectively **transfer learned physical relationships to real‑world data**.
  - We clarify that the AR framework is better suited for climate forecasting because it captures **slowly evolving large‑scale processes** and **multi‑scale dependencies** that diffusion models struggle to represent.

- **Problem Reformulation**:

  - We clarify that our framework is primarily designed for **seasonal‑to‑interannual (S2I)**  forecasting.

  - We extend the forecasting horizon to 14‑month horizons in the low latitude region.


We appreciate the reviewers’ encouraging feedbacks and insightful suggestions. **Reviewer C4po**  acknowledged that the addition of CRPS and additional evaluation methods **strengthen the evaluation of the model**. In addition, **Reviewer 36PM is willing to change the score** if we reformulate our framework and evaluation. We believe that the added analyses and revisions directly address all these concerns and significantly strengthen the manuscript.

Detailed revisions, experiments, and discussions are presented in the point-by-point responses below. We thank the reviewers again for their patient reading and valuable suggestions, which greatly helped us to improve our manuscript.

---

### Meta-Review · Area_Chair_LByV · 2026-01-05

**Summary:**

The authors introduces a visual AR model for climate modeling with evaluations on the ERA5 dataset that demonstrate improvements in certain metrics compared to some baselines.
The main concern was:
1. The paper is posed as probabilistic, climate model that can address the shortcomings of current generative models. However, the paper's evaluation and experiments are not comprehensive in this regard (see next Q) even after the rebuttal process.

Other concerns were raised on insufficient ablations, inference cost, physical consistency of the solutions, climate statistics. I believe the authors have mostly addressed these concerns.

**Reviewer Concerns:**

The probabilistic aspect of the model was only partially addressed by the authors:

1. The authors include CRPS. However, other suggested metrics such as spread-skill (not computed) or comprehensive analysis of the PSD seemed a bit incomplete.
2. CRPS is compared against deterministic weather models such as GraphCast and not generative weather models such as GenCast that the authors reference already. Neither is there comparisons with climate models such as ACE or more physically consistent models such as NeuralGCM. Hence, it is difficult to judge the ~20% or so improvements.
3. A review raised a concern of the model focusing more on O(10) month rollout, which makes it more of a seasonal weather model. Again, the rebuttal partially addresses this, but the framing of the paper is still around climate models that focus on multi-decade simulations.
4. Results also seem to be on a 1deg grid on ERA5 (a weather dataset) while the baselines were trained on 0.25deg - there is not much discussion on these differences.
5. (Minor) Variables are unpredictable in the lead times considered. Tab 1. still shows RMSEs that is not very useful. ACC values are also quite small in general due to this, so interpreting the values is a little challenging. While the authors included suggested metrics based on statistics and time averaged quantities, it might have been better to focus more on these as climate models do, in general.

I believe the authors addressed the other concerns fairly satisfactorily for ICLR. Some questions on physical validity did not seem to have fully satisfactory answers, but this may be a bit out of scope.

However, the main concern still seems outstanding and hence I believe the paper is still in borderline.

**Reviewer Scores:**

The scores stand 8,4,4. I believe the scores of 4s would have both risen to borderline accepts.

---

### Decision · Program_Chairs · 2026-01-26

Reject